# The Arabidopsis MIK2 receptor elicits immunity by sensing a conserved signature from phytocytokines and microbes

Shuguo Hou[1,6✉], Derui Liu[2,6], Shijia Huang[3,6], Dexian Luo[2], Zunyong Liu[2], Qingyuan Xiang[4], Ping Wang[2], Ruimin Mu[1], Zhifu Han[3], Sixue Chen[4], Jijie Chai[3,5], Libo Shan[2✉] & Ping He[2✉]

Sessile plants encode a large number of small peptides and cell surface-resident receptor kinases, most of which have unknown functions. Here, we report that the *Arabidopsis* receptor kinase MALE DISCOVERER 1-INTERACTING RECEPTOR-LIKE KINASE 2 (MIK2) recognizes the conserved signature motif of SERINE-RICH ENDOGENOUS PEPTIDEs (SCOOPs) from *Brassicaceae* plants as well as proteins present in fungal *Fusarium* spp. and bacterial *Comamonadaceae*, and elicits various immune responses. SCOOP signature peptides trigger immune responses and altered root development in a MIK2-dependent manner with a sub-nanomolar sensitivity. SCOOP12 directly binds to the extracellular leucine-rich repeat domain of MIK2 in vivo and in vitro, indicating that MIK2 is the receptor of SCOOP peptides. Perception of SCOOP peptides induces the association of MIK2 and the coreceptors SOMATIC EMBRYOGENESIS RECEPTOR KINASE 3 (SERK3) and SERK4 and relays the signaling through the cytosolic receptor-like kinases *BOTRYTIS*-INDUCED KINASE 1 (BIK1) and AVRPPHB SUSCEPTIBLE1 (PBS1)-LIKE 1 (PBL1). Our study identifies a plant receptor that bears a dual role in sensing the conserved peptide motif from phytocytokines and microbial proteins via a convergent signaling relay to ensure a robust immune response.

[1] School of Municipal and Environmental Engineering, Shandong Jianzhu University, Jinan, China. [2] Department of Biochemistry and Biophysics, Texas A&M University, College Station, TX, USA. [3] Innovation Center for Structural Biology, Tsinghua-Peking Joint Center for Life Sciences, School of Life Sciences, Tsinghua University, Beijing, China. [4] Department of Biology, Genetics Institute, Plant Molecular and Cellular Biology Program, University of Florida, Gainesville, FL, USA. [5] Max-Planck Institute for Plant Breeding Research, Institute of Biochemistry, University of Cologne, Cologne, Germany. [6] These authors contributed equally: Shuguo Hou, Derui Liu, Shijia Huang. ✉email: hou_shuguo@126.com; lshan@tamu.edu; pinghe@tamu.edu

Sessile plants have evolved a large number of cell surface-resident receptor proteins to sense and respond to developmental and environmental cues. Many of these receptors are receptor kinases (RKs) with an extracellular domain perceiving the cognate ligands, a transmembrane domain, and a cytoplasmic kinase domain activating downstream signaling[1–4]. Plant RKs are classified into different subfamilies based on their extracellular domains. Leucine-rich repeat-RKs (LRR-RKs) with a few to more than 30 extracellular LRRs represent the largest subfamily RKs in plants[2,5]. LRR-RKs are found to perceive plant growth hormone brassinosteroids (BRs), various endogenous and exogenous peptide ligands involved in plant growth, reproduction, cell differentiation, immunity, and beyond[1,5], hydrogen peroxide[6] and quinone[7]. Upon the ligand perception, LRR-RKs often heterodimerize with SOMATIC EMBRYOGENESIS RECEPTOR-LIKE KINASE (SERK) subfamily LRR-RKs, e.g., BRASSINOSTEROID INSENSITIVE 1 (BRI1)-ASSOCIATED RECEPTOR KINASE 1 (BAK1)/SERK3 and SERK4, leading to the trans-phosphorylation between the cytoplasmic kinase domains of receptors and SERK coreceptors, and the subsequent activation of the cytoplasmic signaling events[8,9].

Some plant LRR-RKs are pattern recognition receptors (PRRs) that recognize pathogen- or microbe-associated molecular patterns (PAMPs or MAMPs) or damage-associated molecular patterns (DAMPs) and initiate pattern-triggered immunity (PTI)[10–13]. For examples, FLAGELLIN-SENSITIVE 2 (FLS2) and ELONGATION FACTOR-TU (EF-Tu) RECEPTOR (EFR) recognize bacterial flagellin and EF-Tu, respectively[14,15]; PEP1 RECEPTOR 1 (PEPR1)/PEPR2 and RLK7 recognize plant endogenous peptides PLANT ELICITOR PEPTIDE 1 (Pep1), and PAMP-INDUCED SECRETED PEPTIDE 1 (PIP1)/PIP2, respectively[16–19]. Despite the different origins, MAMPs and DAMPs activate the conserved downstream signaling events, including cytosolic calcium influx, reactive oxygen species (ROS) burst, mitogen-activated protein kinase (MAPK) activation, ethylene production, transcriptional reprogramming, callose deposition, and stomatal closure to prevent pathogen entry[11,13,20–22]. Accumulating evidence indicates that the immune response triggered by DAMPs is to amplify MAMP-activated defense since peptidic DAMPs are usually induced upon MAMP perception[10,20,21]. Some plant secreted peptides, such as SERINE-RICH ENDOGENOUS PEPTIDE 12 (SCOOP12), have been shown to induce typical MAMP/DAMP-triggered immune response, the receptors of which have not been identified[23].

The LRR-RK MALE DISCOVERER 1 (MDIS1)-INTERACTING RECEPTOR-LIKE KINASE 2 (MIK2) was initially identified as a component of a receptor heteromer involving in guided pollen tube growth by perceiving the female attractant peptide LURE1[24]. However, the physical interaction between MIK2 and LURE1 was not detected[25]; instead, LURE1 bound directly to another LRR-RK PRK6[25], and the pollen-specific PRK6 was shown to be the receptor of LURE1 family peptides biochemically and genetically[25–27]. Thus, the ligand of MIK2 remains enigmatic. In addition, MIK2 plays a role in response to diverse environmental stresses, including cell wall integrity sensing, salt stress tolerance, and resistance to the soil-borne fungal pathogen *Fusarium oxysporum*[28–31]. A recent study suggested that MIK2 functions as a PRR perceiving an unknown peptide elicitor from *F. oxysporum* to activate plant immunity[32]. Here, we provide genetic and biochemical evidence to show that MIK2 is the bona fide receptor of SCOOP family peptides. SCOOP12 directly binds to the extracellular domain of MIK2 and induces MIK2 dimerization with BAK1/SERK4 coreceptors. Interestingly, SCOOP-LIKE signature motifs are also present in the genomes of a wide range of *Fusarium* spp. and some *Comamonadaceae* bacteria.

*Fusarium* SCOOP-LIKE peptides activate the MIK2-BAK1/SERK4-dependent immune responses, pointing to a dual role of MIK2 in the perception of both MAMPs and DAMPs.

## Results

**The kinase domain of MIK2 elicits specific responses.** We have previously identified RLK7 as the receptor of the secreted peptides PIP1 and PIP2[19]. RLK7 belongs to the subfamily XI of LRR-RKs, including PEPRs and MIK2 (Supplementary Fig 1a). Similar to *RLK7* and *PEPR1*, *MIK2* is transcriptionally upregulated upon treatments of flg22, a synthetic 22-amino-acid peptide derived from bacterial flagellin, and elf18, a synthetic 18-amino-acid peptide derived from bacterial EF-Tu (Supplementary Fig. 1a). To explore the specific function of the extracellular and the cytosolic kinase domains of LRR-RKs, we generated a chimeric receptor consisting of the extracellular domain (ECD) of RLK7 and the transmembrane and cytoplasmic kinase domains (TK) of MIK2 ($RLK7^{ECD}$-$MIK2^{TK}$). The chimeric $RLK7^{ECD}$-$MIK2^{TK}$ and the full-length RLK7 ($RLK7^{ECD}$-$RLK7^{TK}$) under the control of the cauliflower mosaic virus (CaMV) *35S* promoter were transformed into the *rlk7* mutant (Fig. 1a).

Compared to the *rlk7* mutant, both *RLK7/rlk7* and $RLK7^{ECD}$-$MIK2^{TK}$/*rlk7* transgenic seedlings restored the sensitivity to PIP1 treatment exhibiting root growth retardation (Fig. 1b, c), indicating that the chimeric $RLK7^{ECD}$-$MIK2^{TK}$ receptor is functional. Notably, $RLK7^{ECD}$-$MIK2^{TK}$/*rlk7* seedlings showed more substantial root growth retardation to PIP1 treatment than *RLK7/rlk7* (Fig. 1b, c). Besides, PIP1 treatment caused brown roots and darkened root-hypocotyl junctions in $RLK7^{ECD}$-$MIK2^{TK}$/*rlk7* seedlings, but not in *RLK7/rlk7* seedlings (Supplementary Fig. 1b). Furthermore, PIP1 treatment triggered a robust production of ROS in $RLK7^{ECD}$-$MIK2^{TK}$/*rlk7* seedlings compared to that in *RLK7/rlk7* seedlings, which weakly induced ROS production (Fig. 1d)[19]. We also generated a chimeric receptor carrying the ectodomain of PEPR1 and the transmembrane and cytoplasmic kinase domains of MIK2 ($PEPR1^{ECD}$-$MIK2^{TK}$) and transformed it into the *pepr1/pepr2* (*pepr1,2*) mutant (Supplementary Fig. 1c). The $PEPR1^{ECD}$-$MIK2^{TK}$/*pepr1,2* transgenic seedlings also showed severe root growth retardation, brown roots, and darkened root-hypocotyl junctions compared to WT seedlings upon Pep1 treatment (Supplementary Fig. 1d, e). Thus, the MIK2 kinase domain likely triggers specific responses differently from those mediated by the RLK7 and PEPR1 kinase domains. However, we cannot completely rule out the possibility that this was attributed to the increased expression of $MIK2^{TK}$ in $RLK7^{ECD}$-$MIK2^{TK}$/*rlk7* and $PEPR1^{ECD}$-$MIK2^{TK}$/*pepr1,2* transgenic plants.

$MIK2^{TK}$-specific responses prompted us to investigate if MIK2, like RLK7 and PEPR1, acts as a receptor to perceive an endogenous peptide ligand(s). DAMP peptide signaling often induces the expression of themselves or precursors, thereby providing positive feedback for immune signal amplification[10,13]. We thus conducted an RNA-sequencing (RNA-Seq) analysis and focused on small peptide genes. Among 51 small peptide genes induced by PIP1 treatment in $RLK7^{ECD}$-$MIK2^{TK}$/*rlk7*, 18 genes encode SCOOPs[23], SECRETED TRANSMEMBRANE PEPTIDEs (STMPs)[33], and SCOOP/STMP-like proteins (Fig. 1e, Supplemental Fig. 2a, Supplementary Data 1). SCOOPs and STMPs have been recently identified as two groups of secreted peptides with roles in plant growth and defense[23,33]. SCOOPs and STMPs share certain degrees of similarities at the amino (N)-terminal signal peptide and/or the carboxyl (C)-terminal domain with overlapping family members, such as SCOOP4/STMP10, SCOOP13/STMP1, and SCOOP14/STMP2 (Supplementary Fig. 2b).

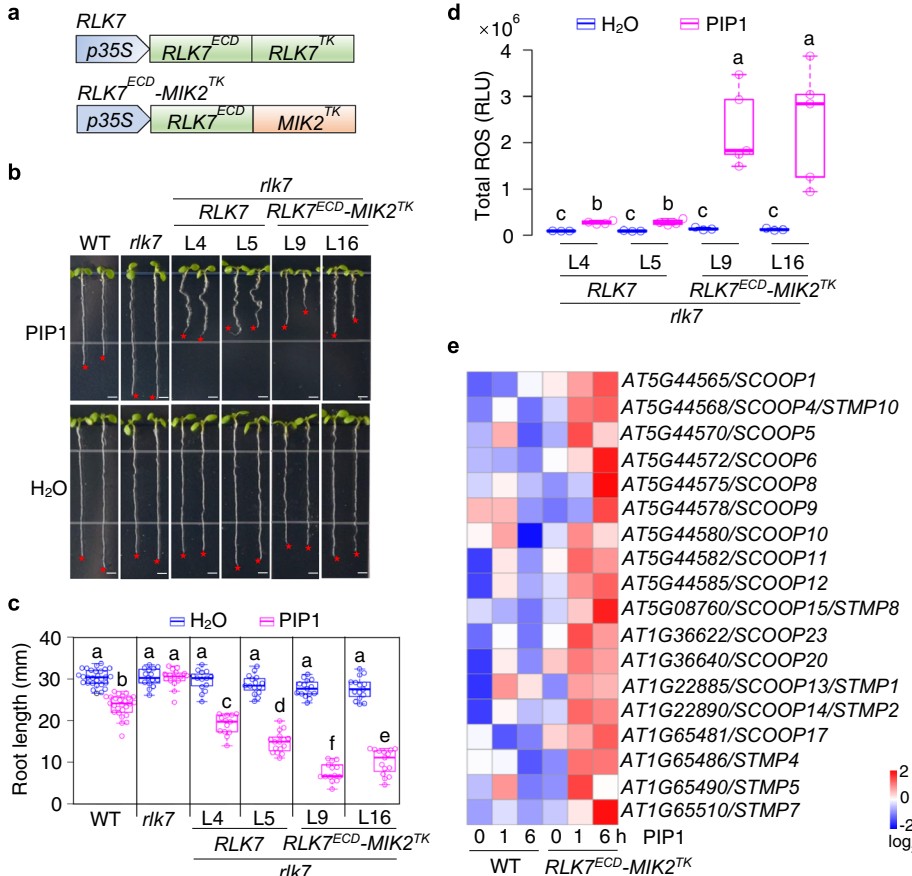

**Fig. 1 Activation of the cytosolic kinase domain of MIK2 induces *SCOOP* expression. a** Diagram of the $RLK7^{ECD}$-$MIK2^{TK}$ chimeric receptor gene. $RLK7^{ECD}$-$MIK2^{TK}$ chimeric gene encoding RLK7 extracellular domain ($RLK7^{ECD}$) and MIK2 transmembrane and cytoplasmic kinase domains ($MIK2^{TK}$) and the full-length RLK7 ($RLK7^{ECD}$-$RLK7^{TK}$) driven by the *CaMV35S* promoter were transformed into *rlk7*. **b**, **c** $RLK7^{ECD}$-$MIK2^{TK}$/*rlk7* transgenic seedlings show more severe root growth inhibition to PIP1 treatment than RLK7/*rlk7* seedlings. Images of 10-day-old seedlings of *Arabidopsis* WT (Col-0), *rlk7*, two representative lines (L9 and L16) of $RLK7^{ECD}$-$MIK2^{TK}$/*rlk7*, and two lines (L4 and L5) of RLK7/*rlk7* grown on ½MS plates with or without 1 µM PIP1: red stars indicate the root tips, scale bar, 4 mm (**b**; quantification of root length of seedlings (**c**)). **d** PIP1 treatment induces a more pronounced ROS production in $RLK7^{ECD}$-$MIK2^{TK}$/*rlk7* than that in RLK7/*rlk7* transgenic seedlings. One-week-old seedlings grown on ½MS plates were treated with $H_2O$ or 100 nM PIP1, and ROS production was calculated as the total relative luminescence units (RLU) (see "Methods" for details). **e** PIP1 treatment induces the expression of *SCOOP* genes in $RLK7^{ECD}$-$MIK2^{TK}$/*rlk7* transgenic plants but not in WT plants. Ten-day-old seedlings grown on ½MS plates were treated with 1 µM PIP1 for 0, 1, and 6 h for RNA-Seq analysis. Heatmap represents adjusted $\log_2$-transformed transcript levels of SCOOPs and STMPs (see "Methods" for details). The box plots in (**c**) and (**d**) show the first and third quartiles as bounds of box, split by the medians (lines), with whiskers extending 1.5-fold interquartile range beyond the box, and minima and maxima as error bar. Different letters indicate a significant difference with others ($P < 0.05$, one-way ANOVA followed by Tukey's test, $n = 28, 26, 15, 15, 15, 13, 15, 16, 15, 14, 15, 15$ for plots from left to right in (**c**), $n = 3, 4, 3, 4, 3, 5, 3, 5$ for plots from left to right in (**d**)). The experiments in (**b**–**d**) were repeated three times with similar results.

**Multiple SCOOP peptides activate diverse plant immune and physiological responses**. Fourteen *SCOOPs* (*SCOOP1-14*) have been identified in the *Arabidopsis* genome[23]. Some peptide-encoding genes induced in $RLK7^{ECD}$-$MIK2^{TK}$/*rlk7* seedlings upon PIP1 treatment, such as *AT1G36640, AT1G36622, STMP4, 5,* and *7*, bear sequence similarities with *SCOOPs* (Fig. 1e). We thus searched the *Arabidopsis* genome and identified additional nine *SCOOP* family members, namely *SCOOP15-23* (Fig. 2a, Supplementary Fig. 2b). SCOOP15 was also named STMP8[33]. Thus, the *Arabidopsis* genome encodes at least 23 *SCOOP* isoforms characterized by a conserved C-terminus with an SxS motif (where S is serine and x is any amino acid) (Fig. 2a, Supplementary Fig. 2b)[23]. Some SCOOPs, including SCOOP6, 7, 10, 11, and 15, contain two copies (an uppercase A or B was added for each copy) of the conserved domain with an SxS motif (Fig. 2a, Supplementary Fig. 2b). The expression pattern of *SCOOP* genes differs in various plant tissues (Fig. 2b). Of those, *SCOOP10* has the highest expression in mature rosette leaves and shoots of

seedlings. In contrast, *SCOOP12* and *13* have the highest expression in roots and flowers, respectively (Fig. 2b), suggesting that different SCOOPs may bear tissue-specific functions.

SCOOPs possess an N-terminal signal peptide directing peptide secretion, variable regions, and a C-terminal conserved region (Supplementary Fig. 2b). Protein structure prediction indicates that SCOOP12 without the signal peptide forms an α-helix followed by a loop-like structure (Supplementary Fig. 3a). Secreted peptides are typically translated as precursor proteins and subsequently secreted into apoplasts for proteolytic processing[34]. To determine whether SCOOP12 is secreted and processed, we generated *pSCOOP12::SCOOP12-GFP/scoop12* transgenic plants, in which *SCOOP12* fused with *GFP* at the C-terminus under the control of its native promoter was transformed into the *scoop12* mutant. SCOOP12-GFP signals were detected in the plasma membrane as well as apoplasts in leaf cells of transgenic plants (Supplementary Fig. 3b). Because the active SCOOP12 peptide is predicted to localize on SCOOP12

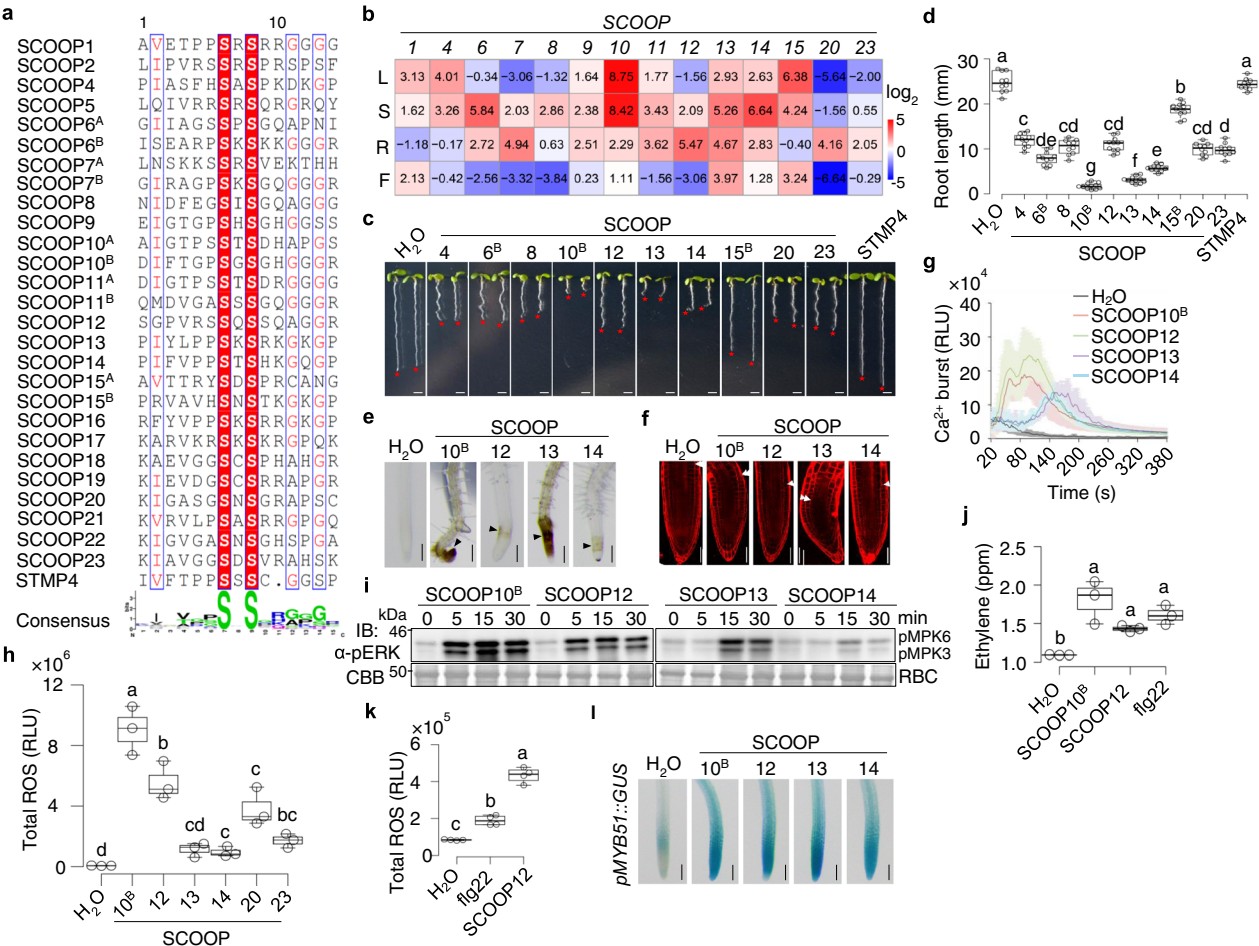

**Fig. 2 SCOOP peptides activate plant immune responses. a** Alignment of SCOOP SxS motifs. Two SxS motifs in SCOOP6, 10, and 11 are respectively indicated with A and B. The conserved and similar residues are in red and yellow, respectively. **b** Expression of *SCOOPs* in different tissues. Heatmap represents adjusted $\log_2$ transformed transcript levels in 4-week-old leaves (L), 7-week-old flowers (F), and 10-day-old shoots (S) and roots (R). Data represent averages of relative expression levels of *SCOOPs* normalized to *UBQ10* from three independent repeats. **c, d** SCOOPs inhibit root growth. Images of 10-day-old seedlings with 1 µM peptides: Red stars indicate the root tips, scale bar, 4 mm (**c**); quantification of root length (**d**). **e** SCOOPs cause root browning. Microscopic images of root tips with arrowheads indicating browning. Scale bar, 500 µm. **f** SCOOPs distort root meristems. Confocal images of root tips of 5-day-old seedlings with 1 µM peptides for 48 h and propidium iodide (PI) staining. Arrowheads indicate boundaries of root meristems and elongation zones. Scale bar, 100 µm. **g** SCOOPs trigger cytosolic $Ca^{2+}$ increases. *p35S::Aequorin* seedlings were treated with 100 nM SCOOPs. Cytosolic $Ca^{2+}$ concentration is indicated as means ± SE of RLU ($n = 4$, SCOOP10[B] and SCOOP12; $n = 3$, $H_2O$, SCOOP13, and SCOOP14). **h** SCOOPs induce ROS production. One-week-old seedlings were treated with 100 nM peptides, and ROS production was calculated as total RLU. **i** SCOOPs induce MAPK activation. Ten-day-old seedlings were treated with 1 µM peptides, and MAPK activation was analyzed by α-pERK1/2 immunoblotting (IB). Coomassie brilliant blue (CBB) staining of Rubisco (RBC) for protein loading. Molecular weight (kDa) was labeled on the left of all immunoblots. **j** SCOOPs induce ethylene production. Ethylene concentration in leaf disks was indicated as parts per million (ppm). **k** SCOOP12 induces a robust ROS production in roots. Detached roots from 1-week-old seedlings were treated with 100 nM peptides. **l** SCOOPs induce expression of *pMYB51::GUS* in roots. Microscopic images of root tips of 1-week-old *pMYB51::GUS* seedlings treated with 1 µM SCOOPs for 3 h and subjected to GUS staining. Scale bar, 1 mm. Box plots in (**d**), (**h**), (**j**), and (**k**) show the first and third quartiles as bounds of box, split by the medians (lines), with whiskers extending 1.5-fold interquartile range beyond the box, and minima and maxima as error bar. Different letters indicate a significant difference with others ($P < 0.05$, one-way ANOVA followed by Tukey's test, $n = 10$, 11, 11, 10, 11, 11, 11,10, 11, 10, 10, 9 for plots from left to right in (**d**); $n = 3$ for plots in (**h**) and (**j**); $n = 4$ for plots in (**k**). Experiments in (**b–e**), and (**g–l**) were repeated three times, and (**f**) twice with similar results.

C-terminus and separates it from two flanking variable regions (Supplementary Fig. 3c), we speculate that SCOOP12 might be proteolytically processed at these sites[34]. In line with our prediction, multiple bands were detected for SCOOP12-GFP in transgenic plants by an immunoblotting analysis (Supplementary Fig. 3d). Together, our data hint that SCOOP12-GFP is likely secreted and proteolytically processed in *Arabidopsis*.

To determine the SCOOP12 processing sites, we followed an established protocol[35,36] and incubated the recombinant HIS-SCOOP12 proteins with the protein extracts isolated from *Arabidopsis* seedlings. Apparently, the HIS-SCOOP12 proteins

were proteolytically processed upon incubation with *Arabidopsis* protein extracts (Supplementary Fig. 3e). The processed products were subjected to a nano-flow liquid chromatography-tandem mass spectrometry (Nano-LC-MS/MS) analysis. A SCOOP12 fragment from leucine 44 ($L^{44}$) to lysine 78 ($Y^{78}$) was identified (Supplementary Fig. 3f), suggesting that SCOOP12 was cleaved after arginine 43 ($R^{43}$) in a dibasic cleavage site (Supplementary Fig. 3g). Importantly, when the dibasic residues of $R^{42}R^{43}$ were substituted with alanine residues ($A^{42}A^{43}$), one of the processed bands of SCOOP12-GFP expressed in protoplasts was abolished (Supplementary Fig. 3h), further implicating the cleavage of

SCOOP12 after R[43]. The data are consistent with the predicted cleavage site between the variable region I and the conserved domain (Supplementary Fig. 3c). We noted that the predicted cleavage site derived from the in vitro assay using plant extracts might not be fully representative of the situation in vivo.

SCOOP12 has been shown to regulate defense response and root elongation in *Arabidopsis*[23]. We tested whether other SCOOPs also have the activity to induce immune-related responses in *Arabidopsis*. Based on the sequence characteristics and expression patterns, we synthesized ten SCOOP peptides (SCOOP4, 6[B], 8, 10[B], 12, 13, 14, 15[B], 20, and 23) corresponding to the SxS motif. We also synthesized the STMP4 peptide, which has an amino acid deletion in the conserved SxS motif. Treatment of all synthesized SCOOP peptides, but not STMP4, inhibited root growth to various degrees, with SCOOP10[B] and 13 being the most active (Fig. 2c, d). A close-up view indicated that prolonged treatments of SCOOP10[B], 12, 13, and 14 peptides in WT seedlings led to brown roots, especially at the root tips of seedlings (Fig. 2e), and distorted root meristems (Fig. 2f), which were not reported for flg22 or Pep1 peptides[16,37]. The browning sites are different in $35S::RLK7^{ECD}$-$MIK2^{TK}$ transgenic plants treated with PIP1 (Supplementary Fig. 1b) and WT plants treated with SCOOPs (Fig. 2e). It is not clear the cause of root browning but could be associated with ROS production and cell wall modifications, such as lignification and suberification, upon SCOOP perception.

Similar to flg22, SCOOP peptides induced various hallmarks of PTI responses, including the cytosolic calcium increase (Fig. 2g), ROS production (Fig. 2h), MAPK activation (Fig. 2i), and ethylene production (Fig. 2j). SCOOP10[B] and 12 exhibited more potent activities than others in triggering PTI responses (Fig. 2g–i). SCOOP13 and 14 had relatively weak activities for $Ca^{2+}$ and ROS bursts but were potent in inhibiting root growth (Fig. 2c, d, g, h). Besides, SCOOP10[B], 12, 13, and 14 triggered the MAPK activation in both shoots and roots (Supplementary Fig. 3i). SCOOP12-induced ROS production in roots was significantly higher than that elicited by flg22 (Fig. 2k). Further, SCOOP10[B], 12, 13, and 14 treatments induced the promoter activities of *MYB51*, a marker gene for root immunity[38], in the roots of *pMYB51::GUS* transgenic plants (Fig. 2l). The data indicate that SCOOPs play a role in immune activation in both shoots and roots.

**MIK2 mediates SCOOP-induced responses**. Next, we tested if MIK2 is required for SCOOP-triggered responses. We isolated two alleles of T-DNA insertional mutants, *mik2-1* (SALK_061769) and *mik2-2* (SALK_046987). Both *mik2-1* and *mik2-2* were insensitive to the treatment of SCOOP10[B] (Fig. 3a, b). The root growth retardation became more pronounced with the increased concentrations of SCOOP10[B] from 1 to 100 nM, which was completely abolished in *mik2-1* (Fig. 3c, d). In addition, the *mik2-1* and *mik2-2* mutants were insensitive to the growth inhibition triggered by the other nine SCOOPs (Fig. 3e, Supplementary Fig. 4a). However, the *mik2-1* mutant responded to the Pep1 treatment similarly to WT plants (Supplementary Fig. 4b). Besides, *MIK2-LIKE* (*MIK2L*), the closest homolog of *MIK2*, and other members of subfamily XI LRR-RKs, including *RLK7*[19], *PEPR1*[17], *PEPR2*[18], *HAESA* (*HAE*), and *HAESA-LIKE2* (*HSL2*)[39], were not involved in responsiveness to SCOOP10[B] (Supplementary Fig. 4c). The *fls2* mutant responded to SCOOP10[B] similarly to WT plants (Supplementary Fig. 4c), supporting that the activities in synthesized SCOOP peptides were unlikely due to the contamination of flg22. The data indicate that the root growth inhibition triggered by SCOOP peptides genetically depends on MIK2, but not other members of the subfamily XI LRR-RKs. Similar to SCOOPs, the apparent MIK2 orthologs with ≥60% of amino acid identity to *Arabidopsis* MIK2 were identified only in *Brassicaceae* plants (Supplementary Fig. 4d)[23,33]. In contrast to the tissue-specific expression of *SCOOPs*, *MIK2* is highly expressed in shoots, roots, and leaves (Supplementary Fig. 4e), consistent with a previous report[28].

We also determined whether SCOOP-mediated other physiological and immune responses depended on MIK2. The SCOOP10[B]- and 12-induced ROS production (Fig. 3f) and MAPK activation (Fig. 3g) were abolished in *mik2* mutants. The SCOOP10[B]- and 12-induced distortion of root meristems was not observed in *mik2-1* (Fig. 3h). RNA-Seq analyses indicate that SCOOP12 treatment for 1 h regulates the expression of 2249 genes (1642 upregulated and 607 downregulated, fold change ≥2 or ≤0.5, p-value ≤0.01) in WT plants compared to 24 regulated genes in *mik2*, suggesting that SCOOP12-regulated gene transcription is almost completely abolished in *mik2* (Supplementary Fig. 4f, Supplementary Data 2). Reverse transcription-quantitative polymerase chain reactions (RT-qPCR) confirmed that the upregulation of three immune-related marker genes, *WRKY30*, *WRKY33*, and *CYP81F2*, by SCOOP10[B], 12, and 13 in WT seedlings was abolished in *mik2* mutants (Fig. 3i). Taken together, these data indicate that SCOOP-triggered responses depend on MIK2.

We next compared genes upregulated by SCOOP12 with genes upregulated by the MIK2 kinase domain in $RLK7^{ECD}$-$MIK2^{TK}$/*rlk7* transgenic plants upon PIP1 treatment (Supplementary Data 2). Interestingly, MIK2 kinase domain-upregulated genes in PIP1-treated $RLK7^{ECD}$-$MIK2^{TK}$/*rlk7* transgenic plants largely overlapped (86.0%, 1036 out of 1205) with SCOOP12-MIK2-upregulated genes compared to its 61.9% (746 out of 1205) overlapping with PIP1-upregulated genes in WT plants (Supplementary Fig. 4g). A heatmap analysis indicates that SCOOP12-regulated genes were clustered together with MIK2 kinase domain-regulated genes in $RLK7^{ECD}$-$MIK2^{TK}$/*rlk7* plants upon PIP1 treatment compared to PIP1-regulated genes in WT (Supplementary Fig. 4h). Similar to PIP1 treatment in $RLK7^{ECD}$-$MIK2^{TK}$/*rlk7* transgenic plants (Fig. 1e), SCOOP12 treatment induced the expression of different *SCOOP* genes in WT plants (Supplementary Fig. 4i). The data further support largely overlapping responses between SCOOP12 perception and MIK2 activation.

**MIK2 is the receptor of SCOOP12**. Since SCOOPs trigger MIK2-dependent responses, we investigated whether SCOOP12 could bind to MIK2. We synthesized the red fluorescent tetramethylrhodamine (TAMRA)-labeled SCOOP12 peptides at its N-terminus (TAMRA-SCOOP12) and determined the ability of WT and *mik2* plants for binding to TAMRA-SCOOP12 in vivo. TAMRA-SCOOP12 peptides were bioactive as they triggered MIK2-dependent root growth inhibition and MAPK activation, despite to a slightly lesser extent than SCOOP12 (Supplementary Fig. 5a–c). Red fluorescent signals were detected in root tips and leaf protoplasts in WT plants, but not in *mik2-1* upon treatment of TAMRA-SCOOP12 (Fig. 4a). Importantly, pretreatment of unlabeled SCOOP12, but not flg22, markedly reduced red fluorescent signals of TAMRA-SCOOP12 in WT seedlings (Fig. 4b), indicating specific and MIK2-dependent binding of SCOOP12 in vivo.

To test whether SCOOP12 directly binds to MIK2 in vitro, we employed isothermal titration calorimetry (ITC) analysis using the extracellular LRR domain of MIK2 (MIK2[ECD]) expressed in insect cells. ITC quantitatively measures the binding equilibrium by determining the thermodynamic properties of protein-protein interaction in solution. As shown in Fig. 4c, MIK2[ECD] bound SCOOP12 potently with a dissociation constant ($K_d$) of 3.18 μM. Alanine (A) substitutions on two conserved serine (S) residues in

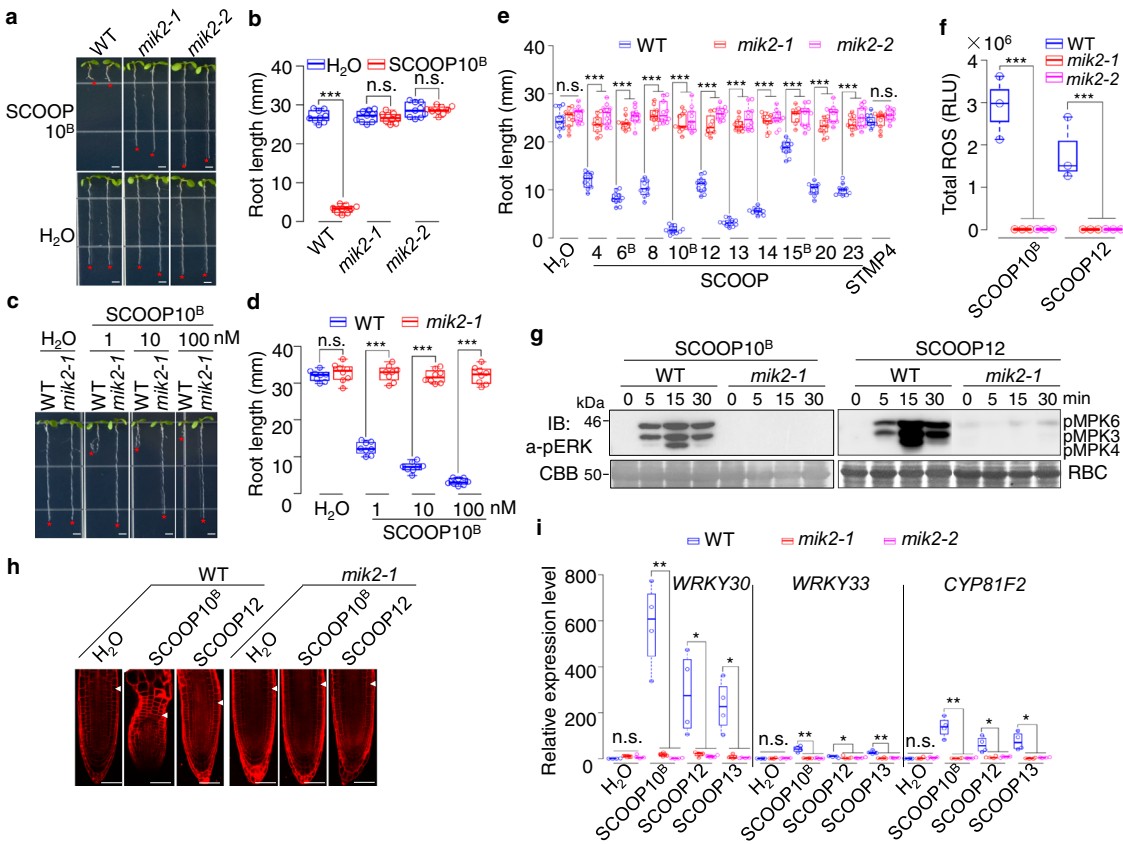

**Fig. 3 SCOOP-triggered responses depend on MIK2. a–d** SCOOP10[B]-triggered root growth inhibition is blocked in *mik2* mutants. Images of 10-day-old WT and *mik2* mutant seedlings with or without 1 μM SCOOP10[B] (**a**) or different concentrations of SCOOP10[B] (**c**). Quantification of root length of seedlings is shown (**b**, **d**) (***$P < 0.001$, n.s., no significant differences, two-tailed Student's *t*-test, $n = 9$ in (**b**), $n = 8$ in (**d**)). Scale bar, 4 mm. **e** The *mik2* mutants block the root growth inhibition induced by different SCOOPs. Quantification of root length of 10-day-old WT and *mik2* seedlings with or without 1 μM SCOOPs (***$P < 0.001$, n.s., no significant differences, two-tailed Student's *t*-test, $n = 9$, plots 1–3, 5, 8–10, 12, 22–24, 27, 29, 31, 32, 36; $n = 10$, plots 4, 6, 7, 11, 15, 16, 18, 19, 21, 25, 30, 33; $n = 8$, plots 14, 17, 20, 26, 34, 35). **f** SCOOP12-induced ROS production is blocked in *mik2* mutants. One-week-old seedlings were treated with 100 nM SCOOP10[B] or SCOOP12, and ROS production was calculated as total RLU (***$P < 0.001$, two-tailed Student's *t*-test, $n = 3$). **g** SCOOP10[B] and SCOOP12-induced MAPK activation is blocked in *mik2-1*. Ten-day-old seedlings were treated with or without 1 μM SCOOP12. The MAPK activation was analyzed by immunoblotting (IB) with α-pERK1/2 antibodies. The protein loading is shown by CBB staining for RBC. **h** The *mik2-1* mutant blocks SCOOP-induced root meristem distortion. Confocal images of root tips of 5-day-old seedlings with or without treatment of 1 μM peptides for 48 h followed by PI staining. Arrowheads indicate the boundary of the root meristem and the elongation zone. Scale bar, 50 μm. **i** SCOOP-induced defense gene expression is blocked in *mik2* mutants. Ten-day-old seedlings were treated with or without 1 μM SCOOP10[B], 12, or 13 for 1 h and subjected to RT-qPCR analysis. Expression of genes was normalized to that of *UBQ10* (*$P < 0.05$, **$P < 0.01$, n.s., no significant differences, two-tailed Student's *t*-test, $n = 4$). The box plots in (**b**), (**d**), (**e**), (**f**), and (**i**) show the first and third quartiles as bounds of box, split by the medians (lines), with whiskers extending 1.5-fold interquartile range beyond the box, and minima and maxima as error bar. The experiments in (**a–d**), (**f**), (**h**), and (**i**) were repeated three times, and (**e**) and (**g**) twice with similar results.

the SxS motif of SCOOP12 (SCOOP12[SS/AA]) abolished its activities to inhibit root growth through MIK2 (Supplementary Fig. 5d, e)[23]. No binding was detected between MIK2[ECD] and SCOOP12[SS/AA] (Fig. 4d). The data indicate that MIK2 recognizes and binds to SCOOP12 directly, and the conserved SxS motif is essential for SCOOP12 binding to MIK2. In addition, we performed surface plasmon resonance (SPR) assays, in which affinity-purified MIK2[ECD] proteins isolated from insect cells were immobilized by an amine-coupling reaction on a sensor chip, and synthesized SCOOP12 peptides were used as the flow-through analytes. The SPR sensorgram showed the profile of SCOOP12 peptides at gradient concentrations flowing through MIK2[ECD] peptides immobilized on the chip (Fig. 4e). The analysis of the binding at equilibrium for SCOOP12 and MIK2[ECD] indicated a calculated $K_d$ of 1.78 μM (Fig. 4f). A similar analysis of the SPR sensorgram using the MIK2[ECD] sensor chip flowing through with SCOOP12[SS/AA] peptides (Fig. 4g) indicated nearly no binding between MIK2[ECD] and SCOOP12[SS/AA] with a $K_d$ of 3481 μM

(Fig. 4h). Together, our data indicate that the extracellular domain of MIK2 directly binds to SCOOP12, and the conserved SxS motif of SCOOP12 is critical for its binding to MIK2. It is noted that the binding constants calculated from the in vitro binding assays are significantly different from the effective concentrations estimated from the physiological experiments. The difference might be due to conditions between in vivo and in vitro assays and the involvement of other cellular components in regulating SCOOP12-MIK2 binding in vivo[40].

### BAK1 and SERK4 are coreceptors of MIK2 in mediating SCOOP-triggered immunity.

Ligand perception by LRR-RK receptors often recruits the BAK1/SERK4 family coreceptors to activate downstream signaling[8,9]. In line with a previous study[23], SCOOP10[B]- and 12-induced root growth inhibition was partially compromised in *bak1-4* compared to WT (Fig. 5a, b). Consistent with the redundant functions of BAK1 and SERK4,

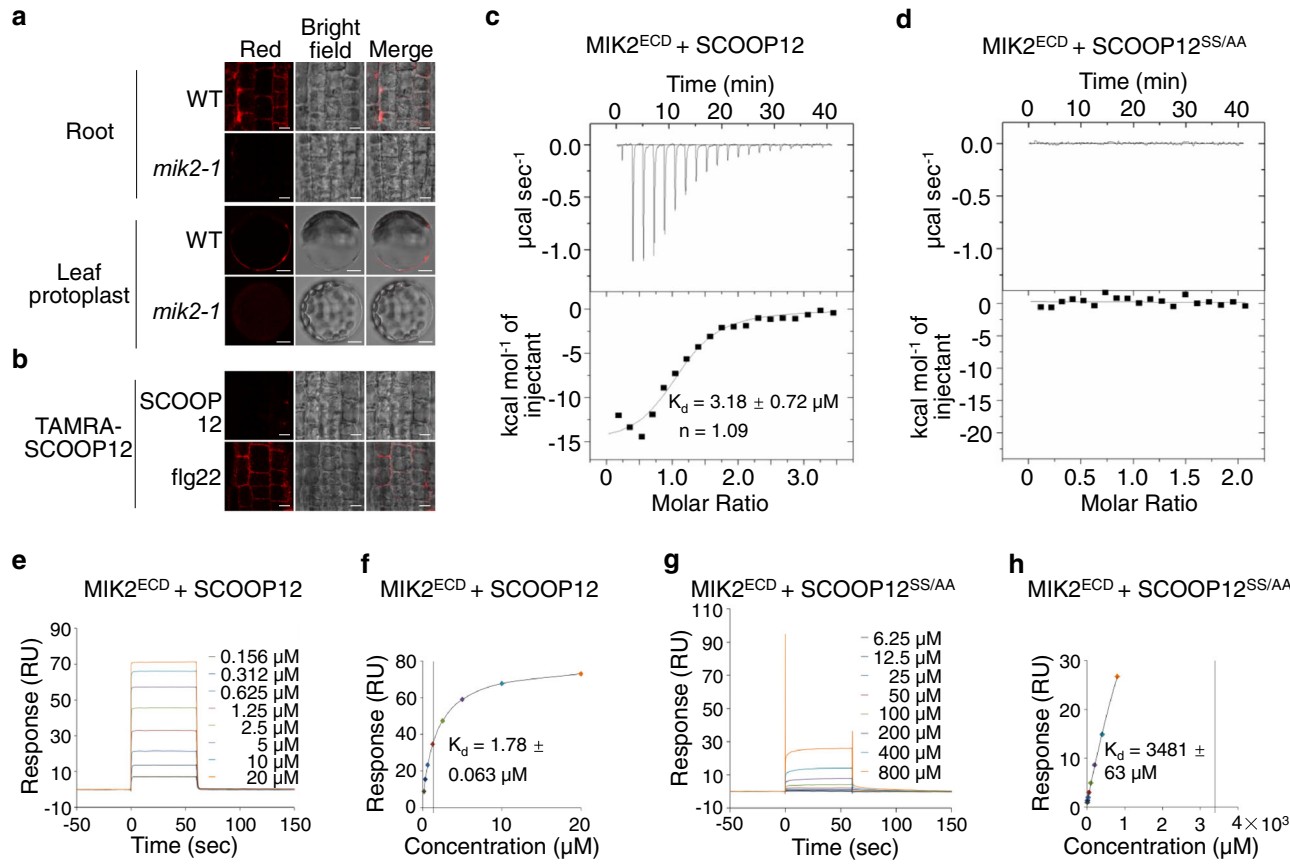

**Fig. 4 SCOOP12 binds to MIK2. a** TAMRA-SCOOP12 peptides label roots and protoplasts from WT but not *mik2-1*. Confocal images of roots or protoplasts of WT and *mik2-1* incubated with 100 nM TAMRA-SCOOP12 (see "Methods" for details). Scale bar, 4 mm (top), 10 μm (bottom). **b** SCOOP12 but not flg22 peptides compete for the binding of TAMRA-SCOOP12 to roots. Confocal images of roots of WT seedlings incubated with 100 nM TAMRA-SCOOP12 in the presence of 1 μM SCOOP12 or flg22 peptides. Scale bar, 4 mm. **c**, **d** SCOOP12 but not SCOOP12$^{SS/AA}$ binds to MIK2$^{ECD}$ with ITC assays. SCOOP12 (**c**) or SCOOP12$^{SS/AA}$ (**d**) was titrated into a solution containing MIK2$^{ECD}$ in ITC cells (see "Methods" for details). The top panels show raw data curves, and the bottom panels show the fitted integrated ITC data curve. The calculated binding kinetic dissociation constant ($K_d$ values ± fitting errors) for SCOOP12 with MIK2$^{ECD}$ is 3.18 ± 0.72 μM, and the stoichiometry of binding (n) is approximately equal to one ligand molecule per receptor molecule (**c**). No binding was detected for SCOOP12$^{SS/AA}$ with MIK2$^{ECD}$ (**d**). **e**, **f** SCOOP12 binds to MIK2$^{ECD}$ with SPR assays. MIK2$^{ECD}$ proteins were immobilized on a sensor chip, and SCOOP12 peptides were used as flow-through analyte for SPR assays (see "Methods" for details). **e** The SPR sensorgram profile of SCOOP12 peptides at gradient concentrations flowing through the MIK2$^{ECD}$ immobilized chip. **f** The steady-state affinity (binding at equilibrium) indicated by a calculated $K_d$ of 1.78 μM. **g**, **h** SCOOP12$^{SS/AA}$ does not bind to MIK2$^{ECD}$ with SPR assays. Similar assays using SCOOP12$^{SS/AA}$ peptides were performed as in (**e**) and (**f**). The SPR sensorgram profile (**g**) indicates nearly no binding between MIK2$^{ECD}$ and SCOOP12$^{SS/AA}$ with a $K_d$ of 3481 μM (**h**). The above experiments were repeated twice with similar results.

the SCOOP-mediated root growth inhibition was alleviated in the *bak1-5/serk4* mutant (Fig. 5a, b). Similarly, SCOOP12-induced ROS production was reduced in *bak1-4* and abolished in *bak1-5/serk4* (Fig. 5c). In addition, SCOOP12- or 13-induced MAPK activation was reduced in *bak1-4* and substantially blocked in *bak1-5/serk4* (Fig. 5d, Supplementary Fig. 6a). To examine whether the SCOOP perception induces MIK2 and BAK1 association, we performed co-immunoprecipitation (co-IP) assays in the *mik2-1* mutant transformed with *GFP*-tagged *MIK2* under the control of its native promoter (*pMIK2::MIK2-GFP/mik2*) with α-BAK1 antibodies. The association between MIK2 and BAK1 was barely detectable in the absence of peptide treatment but was stimulated upon the treatment of SCOOP10$^B$ or 12 (Fig. 5e). SCOOP10$^B$- and 12-induced MIK2-BAK1 association was also observed in protoplasts co-expressing the hemagglutinin (HA)-tagged MIK2 (MIK2-HA) and FLAG epitope-tagged BAK1 (BAK1-FLAG) (Supplementary Fig. 6b). SERK4 was also associated with MIK2 after SCOOP peptide treatments (Fig. 5f).

To test whether the extracellular domains of MIK2 and BAK1 are sufficient to form a SCOOP12-induced complex in vitro, we performed a gel filtration assay of BAK1$^{ECD}$ and MIK2$^{ECD}$ purified from insect cells in the presence of SCOOP12 or SCOOP12$^{SS/AA}$ peptides. The results show that MIK2$^{ECD}$ and BAK1$^{ECD}$ proteins co-migrated in the presence of SCOOP12 (Fig. 5g, h), indicating that SCOOP12 induces the dimerization of MIK2$^{ECD}$ and BAK1$^{ECD}$. The protein complex was eluted mainly at the position corresponding to a size of a monomeric MIK2$^{ECD}$-BAK1$^{ECD}$ complex (~120 kDa) (Fig. 5g, h), suggesting that SCOOP12 may not induce the homodimerization of the MIK2$^{ECD}$-BAK1$^{ECD}$ complex. In contrast, SCOOP12$^{SS/AA}$, which cannot bind to MIK2$^{ECD}$, was unable to induce the dimerization between MIK2$^{ECD}$ and BAK1$^{ECD}$ (Fig. 5i, j), indicating that the SCOOP12-MIK2 binding is required for the MIK2-BAK1 interaction.

**SCOOP-MIK2-BAK1 relays the signaling through the BIK1 family RLCKs.** MAMP/DAMP perception activates receptor-like cytoplasmic kinases (RLCKs), including *BOTRYTIS*-INDUCED KINASE 1 (BIK1) and its close homolog AVRPPHB SUSCEPTIBLE1 (PBS1)-LIKE 1 (PBL1)[41,42]. SCOOP10$^B$ or 12 treatments led to

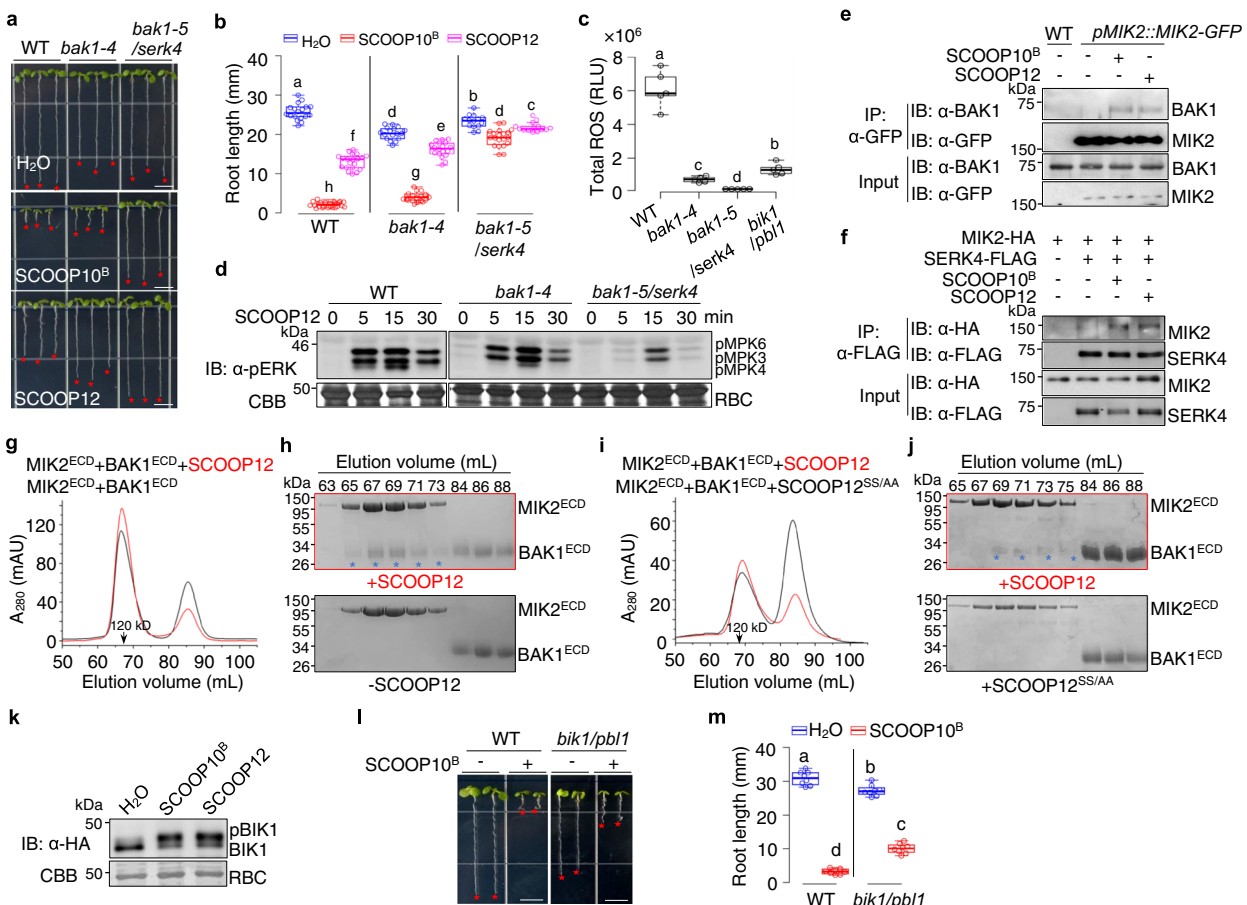

**Fig. 5 SCOOPs induce MIK2-BAK1/SERK4 complex formation and relay signaling through BIK1 and PBL1. a, b** SCOOP-triggered root growth inhibition is compromised in *bak1-4* and *bak1-5/serk4*. Images of 10-day-old seedlings with or without 1 μM SCOOP10[B] or SCOOP12. Scale bar, 4 mm (**a**). Quantification of root length (**b**). **c** SCOOP12-induced ROS is compromised in *bak1-4*, *bak1-5/serk4*, and *bik1/pbl1*. One-week-old seedlings were treated with 100 nM SCOOP12, and ROS production was calculated as total RLUs. **d** SCOOP12-induced MAPK activation is compromised in *bak1-4* and *bak1-5/serk4*. The assay was done as in Fig. 3g. **e** SCOOPs induce MIK2-BAK1 association. Leaves of *pMIK2::MIK2-GFP/mik2-1* plants were treated with or without 1 μM SCOOPs for 30 min. Total proteins were subjected for immunoprecipitation (IP) with α-GFP agarose beads, and the immunoprecipitated proteins were detected with α-BAK1 or α-GFP antibodies (top two). The input controls before IP are shown on the bottom two panels. **f** SCOOPs induce MIK2-SERK4 association. Protoplasts expressing MIK2-HA and SERK4-FLAG, or a control vector (Ctrl) were treated with or without 1 μM SCOOPs for 15 min. IP and IB were performed as (**e**). **g, h** SCOOP12 induces MIK2[ECD]-BAK1[ECD] interaction. Gel filtration chromatography analysis using MIK2[ECD] and BAK1[ECD] shows the elution profiles in the presence (red) or absence (black) of 0.1 mg SCOOP12 (**g**). SDS-PAGE and CBB staining show the eluted fractions (top panel with SCOOP12; bottom panel without SCOOP12). Stars indicate the eluted BAK1[ECD] complexing with MIK2[ECD] (**h**). **i, j** SCOOP12[SS/AA] does not induce MIK2[ECD]-BAK1[ECD] interaction. Experiments were performed as in (**g**) and (**h**). **k** SCOOPs induce BIK1 phosphorylation. Protoplasts expressing BIK1-HA were treated with 1 μM SCOOP10[B] or SCOOP12 for 15 min. Proteins were subjected to IB using α-HA antibodies (top), and CBB staining for RBC as loading controls (bottom). Phosphorylated BIK1 (pBIK1) was indicated as a band mobility shift. **l, m** SCOOP10[B]-triggered root growth inhibition is compromised in *bik1/pbl1*. Images of 10-day-old seedlings with or without 1 μM SCOOP10[B]. Scale bar, 4 mm (**l**). Quantification of root length (**m**). The box plots in (**b**), (**c**), and (**m**) show the first and third quartiles as bounds of box, split by the medians (lines), with whiskers extending 1.5-fold interquartile range beyond the box, and minima and maxima as error bar. A significant difference is shown between different letters (*P* < 0.05, one-way ANOVA followed by Tukey's test, *n* = 15, **b**, 5, **c**, and 8, **m**). Experiments in (**a–d**), (**l**), and (**m**) were repeated three times and (**e–k**) twice with similar results.

the phosphorylation of BIK1, as indicated by a protein band mobility shift in the immunoblot (Fig. 5k). SCOOP12-induced ROS burst and SCOOP10[B]-induced growth inhibition were significantly compromised in the *bik1/pbl1* mutant (Fig. 5c, l, m). Yet, the *bik1/pbl1* mutant did not completely block SCOOP response, likely due to the functional redundancy of RLCKs[41,42] or the existence of an RLCK-independent pathway downstream of SCOOP-MIK2. Nevertheless, the MIK2-BAK1 receptor complex activation by SCOOPs triggers phosphorylation of BIK1 and PBL1 in transducing signaling to downstream events. The activation of NADPH oxidase RBOHD by BIK1 is required for PRR-induced ROS production[43,44]. Likewise, SCOOP12-induced ROS production was abolished in the *rbohD* mutant (Supplementary

Fig. 6c). BAK1 and BIK1/PBL1 are also required for the MIK2-dependent response triggered by a *Fusarium* elicitor[32]. Collectively, our results indicate that the SCOOP-MIK2-BAK1 receptorsome activates the conserved signaling pathways shared by other LRR-RK containing PRR complexes.

**Microbial SCOOP-LIKE peptides trigger a MIK2-dependent immune response.** A recent report showed that the *mik2* mutants were insensitive to an unidentified proteinous elicitor isolated from several *Fusarium* spp., suggesting that MIK2 may perceive a peptide elicitor from *Fusarium*[32]. This prompted us to examine whether SCOOP-LIKE (SCOOPL) peptides are encoded in the

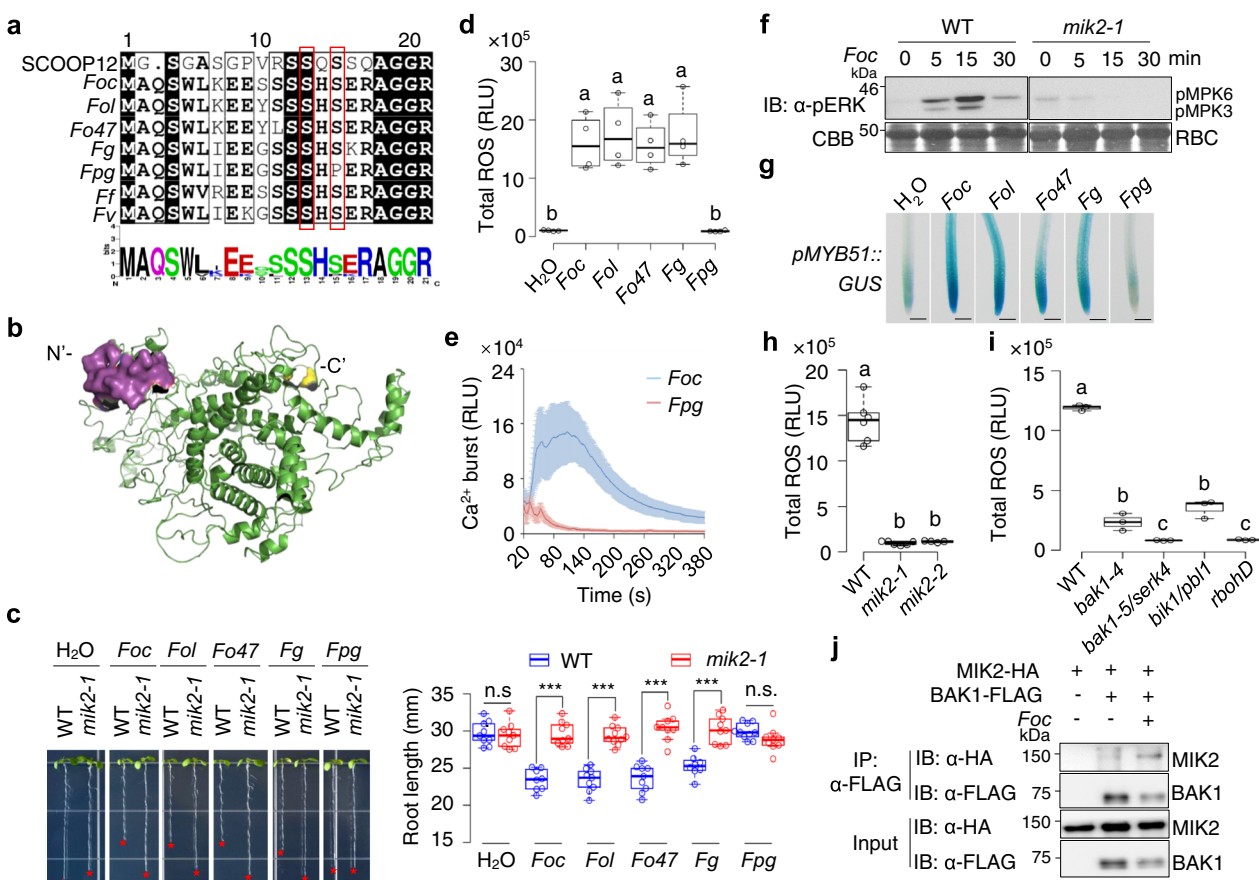

**Fig. 6 Fusarium SCOOP-like peptides activate MIK2-dependent immune responses. a** Alignment of SCOOPL SxS motifs from different Fusarium strains with Arabidopsis SCOOP12. Foc, F. oxysporum f. sp. conglutinans Fo5176; Fol, F. oxysporum f. sp. lycopersici MN25; Fo47, F. oxysporum 47; Fg, F. graminearum PH-1; Fpg, F. pseudograminearum; Ff, F. fujikuroi; Fv, F. venenatum. Two conserved serine (S) residues are boxed in red. **b** The surface accessibility of FgSCOOPL. FgSCOOPL peptide is highlighted in magenta, the C-terminus in yellow, and the rest in green. **c** Fusarium SCOOPL peptides, except FpgSCOOPL, inhibit root growth in a MIK2-dependent manner. Images of 10-day-old WT and mik2-1 seedlings with or without 1 μM Fusarium SCOOPL peptides. Scale bar, 4 mm (left). Quantification of seedling root length (right) (***$P < 0.001$, n.s., no significant differences, two-tailed Student's t-test, $n = 9, 9, 8, 9, 9, 10,$ 9, 9, 8, 9, 10, 9 for plots from left to right). **d** Fusarium SCOOPLs, except FpgSCOOPL, induce ROS production. WT seedlings were treated with or without 1 μM peptides, and ROS was calculated as total RLUs. Different letters indicate a significant difference with others ($P < 0.05$, one-way ANOVA followed by Tukey's test, $n = 4$). **e** FocSCOOPL triggers the cytosolic Ca²⁺ increase. p35S::Aequorin seedlings were treated with or without 1 μM FocSCOOPL or FpgSCOOPL. Cytosolic Ca²⁺ concentration was indicated as means of RLU. The data are shown as mean ± SE ($n = 9$ and 6 for FocSCOOPL and FpgSCOOPL). **f** FocSCOOPL-induced MAPK activation is blocked in mik2-1. Ten-day-old seedlings were treated with or without 1 μM FocSCOOPL. **g** Fusarium SCOOPLs induce pMYB51::GUS expression in roots. Images of root tips of 1-week-old pMYB51::GUS seedlings upon treatments with or without 1 μM SCOOPLs for 3 h and subjected to GUS staining. Scale bar, 1 mm. **h, i** FocSCOOPL-induced ROS production is blocked in mik2 (**h**), and in bak1-4, bak1-5/serk4, bik1/pbl1, and rbohD mutants (**i**). One-week-old seedlings were treated with 1 μM FocSCOOPL, and ROS production was calculated as total RLUs. Different letters indicate a significant difference with others ($P < 0.05$, one-way ANOVA followed by Tukey's test, $n = 6, 6, 4$ for plots from left to right in (**h**), $n = 3$ in (**i**)). **j** FocSCOOPL induces MIK2-BAK1 association. The experiment was performed as in Fig. 5f. Box plots in **c, d, h**, and **i** show the first and third quartiles as bounds of box, split by the medians (lines), with whiskers extending 1.5-fold interquartile range beyond the box, and minima and maxima as error bar. Experiments in (**c–j**) were repeated three times with similar results.

genomes of Fusarium spp. We blast-searched the genome of F. graminearum, a Fusarium strain with a well-annotated genome sequence[45], using SCOOP10[B] and 12 as queries. A 21-amino-acid peptide sequence in the N-terminus of an uncharacterized protein (FGSG_07177) showed a high similarity to SCOOP12 (Fig. 6a, Supplementary Fig. 7a–c). FGSG_07177 is predicted as a transcription regulator with a GAL4-like DNA-binding domain (Supplementary Fig. 7b). Structural modeling indicates that the SCOOPL motif is located on the surface of FGSG_07177 (Fig. 6b). Blast-searching showed that the SCOOPL sequence is highly conserved among all Fusarium species surveyed, and all searched SCOOPLs localize in the N-terminus of FGSG_07177 orthologs, which appear to be a single copy gene in different Fusarium species (Fig. 6a, Supplementary Fig. 7c). Alignment of 22

SCOOPL sequences in FGSG_07177 orthologs from different Fusarium species revealed seven groups, most of which, except for F. pseudograminearum, contain an SxS motif as Arabidopsis SCOOPs (Fig. 6a, Supplementary Fig. 7c). The second conserved serine residue of F. pseudograminearum SCOOPL (FpgSCOOPL) is substituted by a proline (Fig. 6a, Supplementary Fig. 7c). RT-qPCR analyses indicated that the FGSG_07177 homologous gene (FOXB_11846) in Fo5176, a virulent strain on Arabidopsis and belonging to F. oxysporum f. sp. conglutinans[46], is expressed, and its expression was upregulated at 48 h upon fungal inoculation in Arabidopsis roots (Supplementary Fig. 7d).

To examine whether Fusarium SCOOPL peptides trigger immune responses as Arabidopsis SCOOPs, we synthesized five Fusarium SCOOPL peptides from F. oxysporum f. sp.

*conglutinans* strain *Fo*5176 (*Foc*), *F. oxysporum* f. sp. *Lycopersici* (*Fol*), *F. oxysporum* strain *Fo*47 (*Fo*47), *F. graminearum* (*Fg*), and *F. pseudograminearum* (*Fpg*). Four out of five *Fusarium* SCOOPLs, except *Fpg*SCOOPL, which has a polymorphism in the second conserved serine, were able to induce growth inhibition (Fig. 6c), ROS production (Fig. 6d), $Ca^{2+}$ influx (Fig. 6e, Supplementary Fig. 7e, f), MAPK activation (Fig. 6f), and *MYB51* promoter activation in roots (Fig. 6g). *Fg*SCOOPL and *Foc*SCOOPL induced the cytosolic $Ca^{2+}$ increase at the concentration of 10 nM and 100 nM, respectively (Supplementary Fig. 7e, f). *Fusarium* SCOOPL appeared to trigger a weaker ROS production than *Arabidopsis* SCOOP10$^B$ and 12 (Supplementary Fig. 7g). The *Fusarium* SCOOPL-induced growth inhibition, MAPK activation, and ROS production were abolished in the *mik2* mutants (Fig. 6c, f, h). Moreover, the *Foc*SCOOPL-induced ROS production was compromised in the *bak1-5/serk4*, *bik1/pbl1*, and *rbohD* mutants (Fig. 6i). *Foc*SCOOPL also induced MIK2-BAK1 association in *Arabidopsis* protoplasts (Fig. 6j). Thus, *Fusarium* SCOOPLs trigger similar responses with *Arabidopsis* SCOOPs in a MIK2-dependent manner. We knocked out the *SCOOPL* sequence in *FOXB_11846* of *Fo*5176 to test if the SCOOPL plays a role in its pathogenicity (Supplementary Fig. 8a, b). Two independent *Fo*5176 *SCOOPL* knock-out lines showed elevated virulence in *Arabidopsis* compared to WT *Fo*5176 (Supplementary Fig. 8c, d). In addition, as reported previously[28], the *mik2-1* mutant was markedly more susceptible to *F. oxysporum* infections than WT plants (Supplementary Fig. 8e, f).

Interestingly, a SCOOPL motif also exists in the C-terminus of an unknown protein conserved in different genus or species of the bacterial *Comamonadaceae* family, including *Acidovorax* sp., *Curvibacter* sp., *Comamonas* sp., *Ramlibacter* sp., *Limnohabitans* sp., *Verminephrobacter eiseniae*, *Hydrogenophaga* sp., and *Polaromonas* sp. (Supplementary Fig. 9a). SCOOPLs from a strain of *Curvibacter* sp. (*Cu*) and *Verminephrobacter eiseniae* (*Ve*), but not from *Acidovorax temperans* (*At*) or *Acidovorax avenae* (*Aa*) induced growth inhibition and *MYB51* promoter activation in roots (Supplementary Fig. 9b–d). The inactivity of *At*SCOOPL and *Aa*SCOOPL may be due to amino acid variations in the fourth residue in *At*SCOOPL and in the first conserved serine residue in *Aa*SCOOPL (Supplementary Fig. 9a). The *Cu*SCOOPL- or *Ve*SCOOPL-induced growth inhibition was blocked in *mik2-1* (Supplementary Fig. 9b, c). Thus, SCOOPL signature motifs are highly conserved in different microbes, and they may serve as MAMPs perceived by the plant MIK2 receptor.

## Discussion

The *Arabidopsis* genome encodes more than 1000 putative secreted peptide genes, most of which have unknown functions[34]. Similarly, plants have also evolved a large number of RKs, with only a few having defined ligands and functions[3]. In this study, we report that LRR-RK MIK2 recognizes multiple plant endogenous peptides of SCOOP family members, leading to a series of PTI responses, including cytosolic $Ca^{2+}$ influx, ROS burst, MAPK activation, ethylene production, and defense-related gene expression. SCOOP12 directly binds to the extracellular LRR domain of MIK2 in vivo and in vitro, indicating that MIK2 is a bona fide receptor of SCOOPs. The SxS signature motif is essential for SCOOP functions and MIK2 binding. Perception of SCOOPs by MIK2 induces the heterodimerization of MIK2 with BAK1 and SERK4, and BAK1/SERK4 are required for SCOOP-triggered responses, indicating that BAK1/SERK4 are coreceptors of MIK2 in perceiving SCOOPs. SCOOPs have been identified as secreted peptides specific to the *Brassicaceae* family[23]. Interestingly, the SCOOP active motif was also found in a conserved yet uncharacterized protein ubiquitously present in fungal *Fusarium*

spp. and bacterial *Comamonadaceae*. Both *Fusarium* and *Comamonadaceae* SCOOP-LIKE peptides with the SxS signature activate the MIK2-dependent immune responses. Thus, our data reveal a dual role of MIK2 in perceiving the conserved SCOOP signature motif from plants and microbes in triggering plant immunity (Supplementary Fig. 10a). Our study is consistent with an accompanying report in which MIK2 was identified as a receptor of plant SCOOPs and mediated response to *Fusarium* SCOOP-like peptides[47].

Plant plasma membrane-resident RKs perceive diverse exogenous and endogenous signals via the extracellular domain and activate intracellular responses through the cytoplasmic kinase domain[1–3]. The chimeric receptors with the swapped extracellular and intracellular domains between different RKs have been used to study the specificity of signal perception and signaling activation[48–52]. The extracellular domains of RKs determine the ligand-binding specificity[50,51,53]. The intracellular kinase domains of different RKs also trigger specific responses[48]. For example, the chimeric receptor of the EFR extracellular domain and WALL-ASSOCIATED KINASE 1 (WAK1) cytoplasmic kinase domain triggers defense responses that are activated by oligogalacturonides (OGs), the proposed ligand of WAK1[46]. We show that the chimeric receptor of RLK7$^{ECD}$-MIK2$^{TK}$ or PEPR1$^{ECD}$-MIK2$^{TK}$ activates some immune responses that are not observed upon RLK7 or PEPR1 activation by the corresponding ligands. More importantly, the immune responses triggered by RLK7$^{ECD}$-MIK2$^{TK}$ or PEPR1$^{ECD}$-MIK2$^{TK}$ largely overlapped with those activated by SCOOPs, the ligands of MIK2. For example, RLK7 activation by ligand PIP1 moderately inhibits root growth and weakly induces ROS production[19]. In contrast, the RLK7$^{ECD}$-MIK2$^{TK}$ activation by PIP1 causes severe growth inhibition and robust ROS production, similar to SCOOP treatments. PIP1-activated responses are BIK1-independent[19], whereas SCOOPs induce BIK1 phosphorylation and BIK1-dependent responses. Thus, upon ligand perception by the extracellular domain, the cytosolic kinase domain of RKs activates some convergent and unique signaling events, likely through differential phosphorylation events and interaction with various partners.

Fourteen *SCOOPs* have been previously identified in *Arabidopsis*, and SCOOP12 can activate defense response and regulate root elongation[23]. Our study extends the SCOOP family to 23 members, and all tested ones with the conserved SxS motif activate plant immune response. It remains unknown why plants have evolved so many SCOOPs and the functional specificity for different SCOOPs. Notably, *SCOOP* genes show different expression patterns, with some highly expressed in leaves and some in roots. SCOOP12, the most studied SCOOP, is highly expressed in roots (Fig. 2b), and was indicated to regulate root development and resistance to necrogenic bacterium *Erwinia amylovora* and necrotrophic fungus *Alternaria brassicola*[23]. However, *MIK2* is ubiquitously expressed in different plant tissues (Supplementary Fig. 4e, and http://bar.utoronto.ca/efp/cgi-bin/efpWeb.cgi)[28], and regulates many plant processes, including cell wall damage response, salt tolerance, and pollen tube development[24,28]. It is likely that specific SCOOPs might be recognized by MIK2 in different plant tissues to regulate distinct physiological responses. SCOOP12 triggers a robust immune response in roots, and physiological changes, such as darkened hypocotyl-root junctions and distorted meristems in roots. In addition, the *mik2* mutant showed increased susceptibility to the root-invading pathogen *F. oxysporum* (Supplementary Fig. 8e, f)[28]. However, the enhanced susceptibility to *F. oxysporum* has not been observed in the *scoop12* mutant, likely due to the functional redundancy of multiple *SCOOPs* in *Arabidopsis*. Alternatively, it could be attributed to *SCOOPL* genes in *F. oxysporum*.

SCOOPL sequences are also present in a wide range of fungal *Fusarium* spp. and bacterial *Comamonadaceae*. In *Fusarium* spp., the SCOOPL motif we identified is located in the N-terminus of a highly conserved GAL4 DNA-binding domain-containing protein with uncharacterized functions. It is possible that SCOOPL motifs are also present in some other proteins in *Fusarium* spp[47]. Some of these SCOOPLs are active in triggering MIK2- and BAK1/SERK4-dependent immune responses (Fig. 6c, h, i). Knock-out of the *SCOOPL* in *Fo*5176 enhanced the fungal pathogenicity in *Arabidopsis* (Supplementary Fig. 8c, d). Therefore, *Fusarium* SCOOPLs may be recognized as MAMPs by plants. It will also be interesting to determine whether *Fusarium* SCOOPL is the proteinous elicitor isolated from several *Fusarium* spp., which triggers a MIK2-dependent immunity[32]. It remains unknown whether this protein is secreted and processed during *Fusarium* infection and how MIK2 perceives it. Notably, elf18, an 18-amino acid synthetic peptide, is derived from the conserved bacterial translation elongation factor EF-Tu, which can be detected on the extracellular surface and secretome of bacteria[54–56]. Some SCOOPL peptides from *Fusarium* spp. and *Comamonadaceae* have amino acid variations in the conserved SxS motif and cannot activate the immune response. This is consistent with the observation that the SxS motif is required for plant SCOOP peptide functions and binding to MIK2. The variations may be due to the pathogens' evolutionary pressure to escape from hosts' perception of immune elicitation.

It has been reported that some pathogens and nematodes deploy mimics of plant endogenous peptides, such as CLA-VATA3/ESR (CLE), PLANT PEPTIDE CONTAINING SUL-FATED TYROSINE (PSY), and RAPID ALKALINIZATION FACTOR (RALF), to promote the pathogenicity by hijacking plant peptide-receptor signaling[57–60]. For example, RALF secreted from *F. oxysporum* induces plant receptor FERONIA-dependent extracellular alkalization to favor fungal multiplication[57]. In contrast, similar to plant SCOOPs, microbial SCOOPLs activate the MIK2-dependent immune responses, suggesting that SCOOPLs act as MAMPs rather than virulence factors. Compared to the wide distribution of SCOOPLs in fungal *Fusarium* spp. and bacterial *Comamonadaceae*, plant SCOOPs are only present in *Brassicaceae* plants (Supplementary Fig. 4d)[23,33]. This indicates that plant *SCOOPs* may have evolved later than microbial *SCOOPLs*. In addition, gene duplications of *SCOOPs* are common in *Brassicaceae* species[23], an indicative for highly evolved genes. For example, twelve *SCOOPs* exist as tandem repeats on *Arabidopsis* chromosome 5[23]. Thus, plant SCOOPs may have convergently evolved to mimic microbial SCOOPLs and amplify SCOOPL-triggered immunity. Phylogenetic analysis indicates that peptide motifs of *Arabidopsis* SCOOPs, *Fusarium* and *Comamonadaceae* SCOOPLs might have evolved independently (Supplementary Fig. 10b). Moreover, plant SCOOPs are derived from small peptide precursor proteins, whereas *Fusarium* and *Comamonadaceae* SCOOPLs reside in proteins belonging to distinct families. The divergence of protein families harboring SCOOP/SCOOPL suggests that SCOOPs and SCOOPLs might have evolved convergently but unlikely by horizontal gene transfers[61].

## Methods

**Plant materials and growth conditions**. The *Arabidopsis thaliana* accession Columbia-0 (Col-0) was used as wild type (WT). T-DNA insertion mutants of *mik2-1* (SALK_061769), *mik2-2* (SALK_046987), *mik2-like* (SALK_112341), and *rlk7-3* (SALK_120595) were obtained from the Nottingham Arabidopsis Stock Centre (NASC). The *fls2* (SALK_141277), *bak1-4*, *bik1-1/pbl1-1*, *bak1-5/serk4-1*, *rbohd* mutants, and the transgenic line expressing *p35S::GFP* have been reported previously[19,62,63]. The *hae/hsl2* mutant was kindly provided by Dr. Reidunn B. Aalen (University of Oslo, Norway), the *pepr1-2/pepr2-2* (*pepr1,2*) mutant by Dr. Zhi Qi (Inner Mongolia University, China), the *scoop12* mutant by Dr. Sébastien

Aubourg (INRA, France), a transgenic line expressing *p35S::Aequorin* by Dr. Marc Knight (Durham University, UK), transgenic line expressing *pMYB51::GUS* by Dr. Frederick M. Ausubel (Harvard Medical School, USA). Plants were grown in soil (Metro Mix 366, Sunshine LP5 or Sunshine LC1, Jolly Gardener C/20 or C/GP) in a growth room at 20–23 °C, 50% humidity, and 75–100 $\mu$E m$^{-2}$ s$^{-1}$ light with a 12-h light/12-h dark photoperiod for 4–5 weeks before protoplast isolation, or ethylene measurement. Seedlings used for analyses of root growth inhibition, MAPK activation, ROS production, cytosolic Ca$^{2+}$ concentration increase, gene transcription, and GUS staining were grown on half-strength Murashige and Skoog ($1/2$MS) plates containing 0.5% (w/v) sucrose, 0.75% (w/v) agar, and 2.5 mM MES, pH 5.8, under the same conditions as plants grown in soil.

**Plasmid construction and generation of transgenic plants**. The *BAK1* (AT4G33430), *SERK4* (AT2G13790), *BIK1* (AT2G39660), and *MIK2* (AT4G08850) coding sequences (CDSs) were amplified by PCR from Col-0 cDNA using the primers containing *Bam*HI at the 5′ end and *Stu*I at the 3′ end. The PCR products were digested by *Bam*HI and *Stu*I, followed by ligation into the *pHBT-FLAG* or *pHBT-HA* vector to generate *pHBT-35S::BAK1-FLAG*, *pHBT-35S::SERK4-FLAG*, *pHBT-35S::BIK1-HA*, *pHBT-35S::MIK2-HA*, and *pHBT-35S::MIK2-GFP* constructs[24,41,62]. *MIK2*$^{ECD}$ (residues 1–707) and *BAK1*$^{ECD}$ (residues 1–220) were amplified by PCR from Col-0 cDNA and cloned into *pFastBac-1* vector with a C-terminal 6× HIS tag[25,62]. The fusion protein vectors carrying *MIK2*$^{ECD}$ or *BAK1*$^{ECD}$ were used for insect cell expression. To obtain the binary vector *pCAMBIA1300-35S::RLK7*, a 2907-bp *RLK7* CDS was PCR-amplified from WT cDNA using gene-specific primers with *Kpn*I and *Sal*I at the 5′ and 3′ ends, respectively, followed by *Kpn*I and *Sal*I digestion and ligation into the *pCAM-BIA1300* vector. To generate the binary vector *pCAMBIA1300* carrying the *RLK7*$^{ECD}$-*MIK2*$^{TK}$ chimeric receptor gene, an 1824-bp fragment encoding *RLK7*$^{ECD}$ was PCR-amplified from WT cDNA using a gene-specific forward primer with *Kpn*I at the 5′ end and a gene-specific reverse primer with 8-bp overlapping sequence from the 5′ end of *MIK2*$^{TK}$, and a 1011-bp fragment encoding *MIK2*$^{TK}$ using a gene-specific forward primer with 8-bp overlapping sequence from the 3′ end of *RLK7*$^{ECD}$ and a gene-specific reverse primer with *Sal*I at the 5′ end. *RLK7*$^{ECD}$ was fused with *MIK2*$^{TK}$ and inserted into the *pUC19* vector using in-fusion recombinant enzymes (Clontech). After digestion with *Kpn*I and *Sal*I, the *RLK7*$^{ECD}$-*MIK2*$^{TK}$ chimeric receptor gene was inserted into a *pCAMBIA1300* vector with the *CaMV 35 S* promoter and the HA tag at the 3′ end to generate *pCAMBIA1300-p35S::RLK7*$^{ECD}$-*MIK2*$^{TK}$. A similar strategy was used to generate *pCAMBIA1300-35S::PEPR1*$^{ECD}$-*MIK2*$^{TK}$. A 2307-bp fragment encoding *PEPR1*$^{ECD}$ was amplified for fusion with *MIK2*$^{TK}$. To obtain the *pHBT-35S::MIK2-HA* constructs, the *MIK2* CDS was PCR-amplified from WT cDNA using gene-specific primers with *Bam*HI and *Stu*I at 5′ and 3′ ends, respectively, followed by *Bam*HI and *Stu*I digestion and ligation into the *pHBT* vector with *HA* sequence at the 3′ end. To generate the binary vector *pCAMBIA1300-pMIK2::MIK2-GFP*, a 2000-bp promoter sequence upstream of the start codon of *MIK2* was PCR-amplified and subcloned into *pHBT-35S::MIK2-GFP* between *Xho*I and *Bam*HI sites. The *pMIK2::MIK2-GFP-NOS* fragment was further PCR amplified and inserted into *pCAMBIA1300* between *Eco*RI and *Sal*I using in-fusion recombinant enzymes. To generate the binary vector *pCAMBIA-pSCOOP12::SCOOP12-GFP*, a 2481-bp fragment containing the *SCOOP12* promoter and coding sequence was PCR-amplified using Col-0 genomic DNA as a template. The fragment was further inserted into *Nco*I-digested *pCAMBIA1300-GFP* using in-fusion recombinant enzymes. The primers used for cloning and sequencing were described in Supplementary Data 3, and the Sanger-sequencing verified all insertions in different vectors.

The binary vectors were introduced into *Agrobacterium tumefaciens* strain GV3101 for the floral dipping method-based *Arabidopsis* transformation. Transgenic plants were selected with 20 μg/mL hygromycin B. Multiple transgenic lines in the T$_1$ generation were analyzed by immunoblotting for protein expression. Two lines with a 3:1 segregation ratio for hygromycin resistance in the T$_2$ generation were selected to obtain homozygous seeds.

To generate a vector for recombinant protein expression of SCOOP12 in *Escherichia coli*, a 147-bp fragment encoding SCOOP12 lacking the N-terminal signal peptide was PCR-amplified from *pCAMBIA-pSCOOP12::SCOOP12-GFP* using gene-specific primers with *Bam*HI and *Hin*dIII at 5′ and 3′ ends, respectively, followed by *Bam*HI and *Hin*dIII digestion and ligation into pET28a-Avi-HIS$_6$-SUMO[64] to create pET28a-Avi-HIS$_6$-SUMO-SCOOP12.

To build *pHBT-SCOOP12-GFP* vector, 237-bp SCOOP12 CDS was PCR-amplified from *pCAMBIA-pSCOOP12::SCOOP12-GFP* using gene-specific primers with *Bam*HI and *Stu*I at the 5′ and 3′ ends, followed by *Bam*HI and *Stu*I digestion and ligation into the *pHBT* vector before GFP sequence. The *pHBT-SCOOP12*$^{R42R43/AA}$-*GFP* was generated by Platinum *Pfx* DNA polymerase-mediated site-directed mutagenesis with *pHBT-SCOOP12-GFP* as a template.

**Generation of *FocSCOOPL* deletion mutants**. The *FocSCOOPL* deletion mutants were generated with a homologous recombination method. A gene replacement cassette containing a geneticin resistance gene (*NEO*) and a 1098-bp fragment upstream and a 920-bp fragment downstream of *SCOOPL* in *F. oxysporum* 5176 (*Fo*5176) was generated with PCR amplification. *Fo*5176 protoplasts were isolated via treating fresh mycelia with 2 mg/mL driselase (D9515, Sigma, MO, USA), 20 mg/mL lysozyme (L-040-25, GOLDBIO, USA), and 15 mg/mL cellulose

(200605-02, Yakult, Japan). The gene replacement cassette was introduced into protoplasts using a polyethylene glycol (PEG)-mediated protoplast transformation method[65]. Briefly, the gene replacement cassette was gently mixed with protoplasts in STC solution (1.2 M sorbitol, 10 mM Tris-HCl, pH 7.5, 10 mM CaCl₂) and 20 min later, SPTC solution (1.2 M sorbitol, 10 mM Tris-HCl, pH 7.5, 10 mM CaCl₂, 40% PEG4000) was added to the mixture with STC/SPTC (3:1). The mixture was incubated at room temperature for 30 min and then transferred into 20 mL regeneration medium (1.0 M sucrose, 0.1% yeast extract, and 0.1% tryptone) overnight at 25 °C. The regenerated protoplasts were screened with 100 µg/mL geneticin and verified by PCR assays. Primers used for PCR amplification and identification of deletion mutants were listed in Supplementary Data 3.

**Peptide synthesis**. PIP1, Pep1, flg22, and all non-labeled SCOOPs and SCOOP-LIKE peptides were synthesized at ChinaPeptides (Shanghai, China). TAMRA-labeled SCOOP12 peptides were synthesized at Biomatik (Delaware, USA). The sequences of synthesized peptides were listed in Supplementary Data 4.

**Root growth assay**. Cold stratified seeds were surface-sterilized with 70% (v/v) ethanol for 5 min and were sown on ½MS plates with or without peptides at indicated concentrations. Ten-day-old seedlings grown on plates vertically in a growth chamber were photographed, and the root lengths of seedlings were measured using Image J (http://rsb.info.nih.gov/ij/).

**RNA sequencing (RNA-Seq)**. Ten-day-old WT Col-0, mik2-1, and RLK$^{ECD}$-MIK$^{TK}$ (L16) seedlings grown on ½MS plates were incubated in 1 mL of liquid ½MS medium overnight. Seedlings were then treated with or without 1 µM PIP1 or SCOOP12 for 1 or 6 h and harvested for RNA isolation. Total RNAs (5 µg) from two biological replicates were pooled for cDNA library construction. cDNA library preparation and sequencing were carried out on an Illumina HiSeq 4000 platform with 150-nucleotide pair-end reads in LC-BIO (Hangzhou, China). Total reads were mapped to the Arabidopsis genome (TAIR10; www.arabidopsis.org) with Hisat, and the read counts for every gene were generated with StringTie. The expression levels of genes were calculated by quantifying the cDNA fragments per kilobase of transcript per million fragments mapped (FPKM). Differentially expressed genes (DEGs) between different treatments were defined by fold change of read counts at 1 and 6 h compared to that at 0 h (differential expression analysis using edgeR, fold change ≥ 2 with p-value ≤ 0.01). DEGs were listed in Supplementary Data 2.

**RNA isolation and reverse transcription-quantitative polymerase chain reactions (RT-qPCR) analysis**. For detection of gene expression in Arabidopsis, the total RNA was extracted from 10-day-old seedlings grown on ½MS plates, or from rosette leaves or inflorescences of 4- or 7-week-old soil-grown plants using TRIzol reagent (Invitrogen). For detecting gene expression in F. oxysporum during infections, 3-week-old Arabidopsis seedlings were gently taken out of soil, and roots were inoculated with $1 \times 10^7$/mL F. oxysporum spores after washing with sterile water to clear off soil chunks. After incubating on sterile wet papers for 0–48 h, F. oxysporum-infected Arabidopsis roots were collected for total RNA extraction using TRIzol reagent (Invitrogen). One microgram of total RNA was reverse-transcribed to synthesize the first-strand cDNA with M-MuLV Reverse Transcriptases (Thermo Fisher Scientific) and oligo(dT) primers following by RNase-free DNase I (Thermo Fisher Scientific) treatment. RT-qPCR analyses were performed on a QuantStudio™ 3 Real-Time PCR Detection System (Thermo Fisher Scientific) using Faster Universal SYBR® Green Master (Roche) and gene-specific primers. The expression of each gene in Arabidopsis and F. oxysporum was normalized to the expression of Arabidopsis UBQ10 and F. oxysporum GAPDH, respectively. The primers used for RT-qPCR were listed in Supplementary Data 3.

**ROS assay**. ROS burst was determined by a luminol-based assay. Five one-week-old seedlings grown on ½MS plates were incubated in 200 µL ddH₂O overnight in a 1.5 mL centrifuge tube. Then, ddH₂O was replaced by 200 µL of reaction solution containing 50 µM of luminol, and 10 µg/mL of horseradish peroxidase (Sigma-Aldrich) supplemented with or without 100 nM or 1 µM peptide. Luminescence was measured immediately after adding the solution with a luminometer (Glomax 20/20n, Promega) with a one-second interval for 15 min. The total values of ROS production were indicated as means of the relative light units (RLUs).

**Measurement of cytosolic Ca²⁺ concentration**. One-week-old seedlings expressing p35S::Aequorin grown vertically on ½MS plates were put into a 1.5 mL centrifuge tube containing 200 µL solution with 1 mM KCl and 1 mM CaCl₂. Aequorin was reconstituted by treating the seedlings with coelenterazine-h (Promega, Beijing, China) in the dark overnight at a final concentration of 10 µM. Luminescence was measured with a luminometer (Glomax 20/20n, Promega) with a one-second interval for 10 min. The values for cytosolic Ca²⁺ concentrations were indicated as means of RLUs.

**Measurement of ethylene production**. Twelve leaves of 4-week-old plants grown on soil were excised into leaf disks of 0.25 cm², followed by overnight incubation in

a 25 mL glass vial with 2 mL of ddH₂O for recovery. Then, ddH₂O was replaced by 1 mL of peptides at 1 µM, and the vials were capped immediately with a rubber stopper and incubated at 23 °C with gentle agitation for 4 h. One mL of the vial headspace was injected into a TRACE™ 1310 Gas Chromatograph (Thermo Fisher Scientific) with FID supported by Chromeleon 7 for quantitation.

**Histochemical detection of GUS activity**. Two-week-old seedlings grown on ½MS plates were immersed and vacuumed in the GUS staining solution (10 mM EDTA, 0.01% [v/v] Silwet L-77, 2 mM potassium ferricyanide, 2 mM potassium ferrocyanide, and 2 mM 5-Bromo-4-chloro-3-indolyl-β-D-glucuronic acid in 50 mM sodium phosphate buffer, pH 7.0) for 5 min, followed by incubation at 37 °C for 12–24 h. Stained samples were fixed with a 3:1 ethanol:acetic acid solution overnight and cleared by lactic acid and photographed with the Olympus SZX10 stereoscopic microscope equipped with DP27 camera.

**Microscopy assays**. For the observation of propidium iodide (PI) stained roots, 5-day-old seedlings grown on ½MS plates were mounted in ddH₂O containing 10 µM PI for 20 min before imaged with the Leica SP8 confocal microscope. Images were captured at 543 nm laser excitation and 578–700 nm emission. For the observation of TAMRA-SCOOP12 labeled roots and protoplasts, 5-day-old seedlings grown on ½MS plates or protoplasts isolated from leaves of 4-week-old soil-grown plants were treated with 100 nM TAMRA-SCOOP12 with or without 1 µM SCOOP12 or flg22 for 5 min in liquid ½MS or W5 solution (154 mM NaCl, 125 mM CaCl₂, 5 mM KCl, 2 mM MES, pH 5.7), followed by washing with ½MS or W5 solution for three times before imaged with the Leica SP8 confocal microscope. Images were captured at 552 nm laser excitation and 570–620 nm emission. To observe SCOOP12-GFP in Arabidopsis pSCOOP12::SCOOP12-GFP/scoop12 transgenic plants, detached leaves were imaged under the Leica SP8 confocal microscope directly or 3 min after 5% sodium chloride (NaCl) treatment. Images were captured at 472 nm laser excitation and 493–547 nm emission. For all the observations, the pinhole was set at one Airy unit, and the imaging processing was carried out by using the Leica Application Suite X (LAS X) software.

**MAPK assay**. Ten-day-old seedlings grown vertically on ½MS plates were transferred to ddH₂O overnight for recovery. Then, ddH₂O was replaced by 100 nM peptides for the indicated time. Each sample containing three seedlings was grounded in 40 µL of extraction buffer (150 mM NaCl, 50 mM Tris-HCl, pH 7.5, 5 mM EDTA, 1% [v/v] Triton X-100, 1 mM Na₃VO₄, 1 mM NaF, 1 mM DTT, 1:200 complete protease inhibitor cocktail from Sigma). The supernatant was collected after 13,000 × g centrifugation for 5 min at 4 °C and protein samples with 1× SDS buffer were loaded on 10% (v/v) SDS-PAGE gel before transfer to a PVDF membrane, which was then blotted using α-pERK1/2 antibodies (1:1500; Cell Signaling, Cat # 9101) for the detection of phosphorylated MPK3, MPK4, and MPK6.

**BIK1 phosphorylation assay**. WT protoplasts were transfected with pHBT-35S::BIK1-HA for 12 h followed by treatment with 1 µM SCOOP peptides for 15 min. Total proteins were isolated with extraction buffer (150 mM NaCl, 50 mM Tris-HCl, pH 7.5, 5 mM EDTA, 1% [v/v] Triton X-100, 1 mM Na₃VO₄, 1 mM NaF, 1 mM DTT, 1:200 complete protease inhibitor cocktail from Sigma). The supernatant collected after 13,000 × g centrifugation for 5 min at 4 °C was loaded on 10% SDS-PAGE gel before transfer to a PVDF membrane, which was then blotted with α-HA antibodies (1:3000; Roche, Cat # 12013819001).

**Co-immunoprecipitation (co-IP) assay and detection of SCOOP12-GFP proteins**. For co-IP assays in protoplasts, protoplasts were co-transfected with pHBT-35S::MIK2-HA and pHBT-35S::BAK1-FALG or pHBT-35S::SERK4-FLAG at 50 µg DNA for 500 µL protoplasts at the density of $2 \times 10^5$/mL for each sample and were incubated for 12 h. After treatment with 1 µM SCOOP peptides for 15 min, samples were collected by centrifugation at 200 × g for 2 min and lysed in 300 µL IP buffer (20 mM Tris-HCl, pH 7.5, 100 mM NaCl, 1 mM EDTA, 2 mM DTT, 10% [v/v] Glycerol, 0.5% [v/v] Triton X-100 and protease inhibitor cocktail from Sigma) by vortexing. After centrifugation at 10,000 × g for 10 min at 4 °C, 30 µL of supernatant was collected for input controls. The remaining supernatant was pre-incubated with protein-G-agarose beads at 4 °C for 1 h with gentle shaking at 100 × g on a rocker. IP was carried out with α-FLAG agarose for 3 h at 4 °C. Beads were collected by 500 × g centrifugation for 2 min and washed three times with washing buffer (20 mM Tris-HCl, pH 7.5, 100 mM NaCl, 1 mM EDTA, 0.1% [v/v] Triton X-100) and once with 50 mM Tris-HCl, pH 7.5. Immunoprecipitated proteins and input proteins were analyzed by immunoblotting with α-HA (1:3000; Roche, Cat # 12013819001) or α-FLAG antibodies (1:3000; Sigma-Aldrich, Cat # A8592).

For co-IP assays in transgenic seedlings, leaves of 4-week-old pMIK2::MIK2-GFP transgenic plants were hand-inoculated with 1 µM SCOOP10$^B$ or SCOOP12 and were treated for 30 min, followed by total protein isolation with extraction buffer (50 mM Tris-HCl, pH 7.5, 150 mM NaCl, 1% [v/v] NP-40 and 1:200 protease inhibitor cocktail from Sigma). Co-IP was carried out the same as for co-IP with protoplasts except that GFP-Trap agarose beads (Chromotek) were used instead. Immunoprecipitated proteins and input proteins were analyzed by

immunoblotting with α-GFP (1:3500; Roche, Cat # 11814460001) or α-BAK1 antibodies (1:2000)[64].

To detect SCOOP12-GFP proteins, five of one-week-old *pSCOOP12::SCOOP12-GFP/scoop12*, *p35S::GFP*, and WT seedlings grown on ½MS plates were harvested and grounded in 40 μL of extraction buffer (150 mM NaCl, 50 mM Tris-HCl, pH 7.5, 5 mM EDTA, 1% [v/v] Triton X-100, 1 mM Na$_3$VO$_4$, 1 mM NaF, 1 mM DTT, 1:200 complete protease inhibitor cocktail from Sigma). The supernatant was collected after 13,000 × *g* centrifugation for 5 min at 4 °C, and protein samples were separated on 15% (v/v) SDS-PAGE and subjected to immunoblotting (IB) with α-GFP antibodies (1:3500; Roche, Cat #11814460001).

**In vitro protein cleavage assay.** Recombinant HIS-SCOOP12 proteins were expressed in *E. coli* BL21 strain with an isopropyl β-D-1-thiogalactopyranoside (IPTG)-inducible system and purified with Ni-NTA agarose (QIAGEN, #166048536) in buffer containing 50 mM Tris-HCl pH 7.5 and 300 mM NaCl. For in vitro protein cleavage assay, 5 μg of purified HIS-SCOOP12 proteins were incubated with 10 μL protein extracts isolated from 10-day-old *Arabidopsis* WT seedlings or protein extraction buffer (50 mM HEPES, pH 7.4, 10 mM EDTA, and 1× protease inhibitor cocktail, Sigma) following a published protocol[36] for 5 h at room temperature with rotation. The reaction was quenched by adding SDS loading buffer followed by separation on 15% (v/v) SDS-PAGE and subjected to immunoblotting (IB) with α-HIS antibodies (1: 2000; Roche, Cat #11965085001) or Coomassie Brilliant Blue (CBB) staining. The processed SCOOP12 bands were cut from SDS-PAGE for LC-MS/MS analysis.

**LC-MS/MS analysis.** The processed SCOOP12 bands were in-gel digested following a published protocol[66], except that digestion was conducted with Lys-C (Promega, Madison, WI, USA). The digested peptides were desalted using micro ZipTip mini-reverse phase (Millipore). In brief, after wetting (with 50% acetonitrile) and equilibrating a ZipTip (with 0.1% formic acid), the peptides were bound to C-18 material. A subsequent washing step with 0.1% formic acid was followed by a final elution with 60% acetonitrile and 0.1% formic acid. The samples were lyophilized to dryness using a labconco speedvac (Centrivap, Labconco Inc., USA). The resulting peptides were resuspended in 0.1% formic acid for mass spectrometric analysis. The mass spectrometry data acquisition was performed on an EASY-nLC 1200 ultraperformance liquid chromatography system (Thermo Scientific Inc., San Jose, CA) connected to an Orbitrap Q-Exactive HF instrument equipped with a nano-electrospray source (Thermo Scientific Inc., San Jose, CA)[67]. The peptide samples were loaded to a C18 trapping column (75 μm i.d. × 2 cm, Acclaim PepMap® 100 particles with 3 μm size and 100 Å pores) and then eluted using a C18 analytical column (75 μm i.d. × 25 cm, 2 μm particles with 100 Å pore size). The flow rate was set at 250 nL/min with solvent A (0.1% formic acid in water) and solvent B (0.1% formic acid and 99.9% acetonitrile) as the mobile phases. MS/MS analysis was carried out in positive mode applying data-dependent acquisition. Survey full-scan MS spectra scan range was 375–1575 *m/z*, with a resolution of 60,000 at 200 *m/z*. The precursor isolation window was set at 4 *m/z*, the automatic gain control (AGC) was 3e6, and the maximum injection time (IT) was 80 ms. The MS/MS resolution was 15,000, and the mass range was 200–2000 *m/z*. The minimum AGC target was 8.0e3, and the IT was 55 ms. The normalized collision energy was set to be 28 with apex trigger of top 7 precursor ions. The tandem mass spectra were extracted from the Xcalibur .raw files and converted into mgf files using Proteome Discoverer 2.4 (Thermo Fisher Scientific Inc., San Jose, CA). All MS/MS samples were analyzed using Mascot Daemon (Matrix Science, London, UK; version 2.4.0). Mascot was set up to search the *Arabidopsis* TAIR10 database assuming the digestion enzyme semi-Lys C. Decoy database searching was enabled. Carbamidomethyl of cysteine was specified in Mascot as a fixed modification. The raw MS/MS spectra were extracted and exported from the raw data file using Xcalibur Qualbrower and annotated manually with the assistance of Mascot and MS Product from ProteinProspector.

**Protein expression and purification.** The ECD domains of MIK2 (MIK2$^{ECD}$) and BAK1 (BAK1$^{ECD}$) fused with a 6 × HIS tag at the C-terminus were expressed using the Bac-to-Bac baculovirus expression system (Invitrogen) in High Five cells at 22 °C[25]. One liter of cells (1.8 × 10$^6$ cells per mL cultured in the medium from Expression Systems) was infected with 20 mL baculovirus and the media were harvested after 48 h of infection. The secreted MIK2$^{ECD}$-6 × HIS and BAK1$^{ECD}$-6 × HIS proteins were purified using Ni-NTA (Novagen) and size-exclusion chromatography (Hiload 200, GE Healthcare) in buffer containing 10 mM Bis-Tris pH 6.0 and 100 mM NaCl.

**Isothermal titration calorimetry (ITC) assays.** The binding affinities of MIK2$^{ECD}$ with SCOOP12 or SCOOP12$^{SS/AA}$ peptides were measured on a MicroCalorimeter ITC200 (Microcal LLC) at 25 °C[25]. All protein samples used in the ITC assay were dialyzed in the buffer containing 200 mM HEPES, pH 7.0, and 100 mM NaCl. Protein concentration was determined by absorbance spectroscopy at 280 nm. SCOOP12 or SCOOP12$^{SS/AA}$ peptides (600 μM) were injected (1.5 μL per injection) into the stirred calorimeter cells containing 30 μM MIK2$^{ECD}$ with 20 × 2 μL at 2.5-min intervals. ITC data were analyzed using MicroCal Origin 7.0.

**Surface plasmon resonance (SPR) assays.** The binding kinetics and affinities of MIK2$^{ECD}$ with SCOOP12 or SCOOP12$^{SS/AA}$ peptides were assessed on a Biacore T200 instrument (GE Healthcare, Pittsburgh, PA) with CM5 sensor chips. The MIK2$^{ECD}$ proteins were exchanged to 10 mM sodium acetate (pH 5.5), and the peptides used for SPR were dissolved in HBS-EP+ (10 mM HEPES, 150 mM NaCl, 3 mM EDTA, and 0.05% [v/v] surfactant P20 (GE Healthcare). About 4700 response units of MIK2$^{ECD}$ proteins were immobilized on a CM5 chip, and a blank channel was used as a negative control. SCOOP12 or SCOOP12$^{SS/AA}$ peptides were diluted into indicated concentrations and injected at a flow rate of 30 μL min$^{-1}$ for 2 min, followed by dissociation for 5 min. After dissociation, 5 mM NaOH was injected for 30 s to remove any non-covalently bound proteins from the chip surface. The binding kinetics was analyzed with the software Biaevaluation® Version 4.1 using the 1:1 Langmuir binding model.

**Gel filtration assays.** The purified MIK2$^{ECD}$ proteins (0.5 mg for each) and BAK1$^{ECD}$ proteins (0.2 mg for each) were incubated together with SCOOP12 or SCOOP12$^{SS/AA}$ (0.1 mg for each) in a buffer containing 10 mM Bis-Tris, pH 6.0, 100 mM NaCl on ice for 1 h. Then, each of the mixtures was separated by gel filtration (Hiload 200, GE Healthcare) with a flow rate of 0.8 mL/min and an injection volume of 40 mL at 4 °C, and the peak fractions were separated on SDS-PAGE followed by Coomassie blue (CBB) staining.

***Fusarium oxysporum* disease assay.** *F. oxysporum* strain *Fo*5176 was grown on potato dextrose agar plates (Difco) for 4 d at 28 °C. The hyphae of *Fo*5176 were inoculated in potato dextrose broth (Difco) and incubated in a shaker (120 rpm) at 28 °C for 5 d. The spores were collected by filtering through Miracloth (Millipore) followed by centrifugation at 3000 × *g* for 15 min, resuspended with sterile water, and adjusted to a final concentration of 10$^7$ spores/mL. Three-week-old *Arabidopsis* seedlings were gently taken out of the soil, and roots were washed with sterile water to clear off soil chunks, blotted using paper towels, immersed in *Fo*5176 spore suspension or water (mock) for 5 min, and then planted back in pots with pre-wet fresh soil. The inoculated seedlings were covered with a dome for two days and incubated at 28 °C, 65% humidity, and 100 μE m$^{-2}$ s$^{-1}$ light with a 12-h light/12-h dark photoperiod. The disease severity was evaluated using the percentage of chlorotic leaves in each plant[68].

**Phylogenetic analysis and graphical illustration.** Protein sequences were retrieved from the NCBI database. Multiple sequence alignments were generated with ClustalW[69] and displayed in ESPript3 (esprit.ibcp.fr). The consensus of sequences was analyzed by WebLogo (http://weblogo.berkeley.edu/logo.cgi). Phylogenetic analysis was carried out using MEGAX with the neighbor-joining method with 1000 bootstrap replicates[70]. The phylogenetic tree was visualized in an interactive tree of life (iTOL, https://itol.embl.de/). Homology modeling and visualization of *F. graminearum* protein FSGG_07177 were performed with Pymol (The PyMOL Molecular Graphics System, Version 1.2r3pre, Schrödinger, LLC.) with the centromere DNA-binding protein complex CBF3 subunit B (PDB-ID, c6f07B) as the template. Three-dimensional structure of SCOOP12 was predicted by I-TASSER (https://zhanglab.ccmb.med.umich.edu/I-TASSER/).

**Statistical analysis.** Statistical analyses were all done in Excel with built-in formulas. The *p* values were calculated with two-tailed unpaired Student's *t*-test analysis for binary comparison or one-way ANOVA and Tukey's post hoc honest significance test to compare more than two genotypes or treatments. The measurements shown in box plots display the first and third quartiles and split by medians (center lines), with whiskers extending to 1.5-fold the interquartile range from the 25th and 75th percentiles. Dots represent outliers. Asterisks illustrate the *p* values: ***$P < 0.001$, **$P < 0.01$, and *$P < 0.05$.

**Reporting summary.** Further information on research design is available in the Nature Research Reporting Summary linked to this article.

## Data availability

The raw RNA-seq data is available in the NCBI database with accession numbers GSE159580 and GSE166301. Source data are provided with this paper.

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

## Acknowledgements
We thank the *Arabidopsis* Biological Resource Center (ABRC) for providing the *Arabidopsis* T-DNA insertion lines, Dr. Reidunn B. Aalen (University of Oslo, Norway), Dr. Zhi Qi (Inner Mongolia University, China), Dr. Frederick M. Ausubel (Harvard Medical School, USA), Dr. Marc Knight (Durham University, UK), and Dr. Sébastien Aubourg (INRA, France) for providing *Arabidopsis* mutants or transgenic lines, Dr. Lijun Ma (University of Massachusetts Amherst, USA) for providing *F. oxysporum* strain Fo5176, Dr. Weicai Yang (IGDB, CAS) for providing *MIK2-HA* construct, and Dr. Pingwei Li (Texas A&M University, USA) for providing pET28a-Avi-His₆-SUMO vector. We also thank Dr. Cyril Zipfel (University of Zurich, Switzerland), and Dr. Ralph Hückelhoven (Technical University of Munich, Germany) for discussions, Dr. Jinrong Xu (Purdue University, USA), and Dr. Huiquan Liu (Northwest A&F University, China) for assisting on the *Fusarium* genome sequence analysis, Dr. Wei Zhang (Shandong University, China) for technique supports, and members of the laboratories of L.S. and P.H. for discussions and comments of the experiments. The work was supported by National Science Foundation (NSF) (IOS-1951094) and NIH (R01GM092893) to P.H., NIH (R01GM097247), the Robert A. Welch Foundation (A-1795) to L.S., Natural Science Foundation of Shandong Province (ZR2020MC022) to S.H., National Natural Science Foundation of China (NSFC) to S.H. (31500971) and Z.H. (31971119), and Youth Innovation Technology Project of Higher School in Shandong Province to S.H. (2020KJF013) and R.M. (2019KJD003).

## Author contributions
S.Hou, D.Liu., P.H., and L.S. conceived, designed experiments, analyzed data, and wrote the manuscript; S.Hou identified SCOOP and SCOOP-LIKE sequences, generated *Arabidopsis* transgenic and *mik2* mutant lines, performed physiological analysis, RNA-Seq and RT-qPCR analyses, SPR assay, ROS production assay, and cytosolic calcium assay; D.Liu performed immunoblotting analysis, protein-protein interaction analysis, ethylene production measurement, microscopy assays, homology modeling, and statistical analysis; S.Huang under the supervision of Z.H. and J.C. purified MIK2^ECD and BAK1^ECD proteins and performed ITC and gel filtration analyses; D.Luo contributed to the root growth and ROS production assays, generated *pMIK2::MIK2-GFP/mik2-1* transgenic lines; Z.L. performed phylogenetic analysis, sequence alignment analysis, the *Fusarium* genome sequence analysis, partial SPR assay, generation of *FocSCOOPL* deletion mutants, and RT-qPCR analysis *of FOXG_11846* expression in *F. oxysporum* during infection; P.W. performed *F. oxysporum* disease assays; Q.X. and S.C. performed LC-MS/MS analysis; R.M., Z.H., and J.C. analyzed the data.

## Competing interests
The authors declare no competing interests.
