## [Peer Review File · Nature Communications]

Immune elicitation by sensing the conserved signature from phytocytokines and microbes via Arabidopsis MIK2 receptorREVIEWER COMMENTS

Reviewer #1 (Remarks to the Author):

Hou et al. reported that SCOOP peptides trigger diverse immune responses via leucine rich-repeat receptor kinase MIK2. They first observed that, when the chimeric RLK7(ECD)-MIK2(TK) receptor was overexpressed in the *rlk7* mutant, SCOOP and STMP peptides were induced after PIP1 treatment. Then they analyzed SCOOP peptides and MIK2 receptor in detail and proposed that SCOOP and MIK2 function as ligand-receptor pair involved in immune responses. The *mik2-1* and *mik2-2* mutants were insensitive to SCOOP peptides, and SCOOP peptide directly binds extracellular LRR domain of MIK2 with a dissociation constant (Kd) of approximately 1-3 μ M. The authors also showed that BAK1 and SERK4 are co-receptors for MIK2 in mediating SCOOP-triggered immunity. In addition, the MIK2-BAK1 receptor complex activation by SCOOPs triggers phosphorylation of receptor-like cytoplasmic kinases BIK1 and PBL1. Interestingly, SCOOP-like motifs are highly conserved in various microbes, and they may serve as MAMPs perceived by the plant MIK2 receptor.

The data was clearly presented, the manuscript was well written, and the take-home message of the work is straightforward. Here are some comments.

- (1) The authors reported that, when the chimeric RLK7(ECD)-MIK2(TK) receptor was overexpressed in the *rlk7* mutant, SCOOP and STMP peptides were induced after PIP1 treatment. This should be important data that inspired the authors to focus on SCOOP peptides, but there is no explanation for the rank of SCOOP genes among all differently expressed genes. Many readers will be surprised at the sudden emergence of the SCOOP genes at this section.
- (2) SCOOP10B is active even at 1 nM but binding constant between SCOOP and MIK2 is approximately 1-3 μ M. The authors need to elaborate on this discrepancy.
- (3) Is there any information about the in vivo natural structure of SCOOP peptides? All experiments are performed on synthetic peptides based on the predicted putative structure.
- (4) Is *mik2* mutant vulnerable to infection with fungal *Fusarium* spp. or bacterial Comamonadaceae?

Reviewer #2 (Remarks to the Author):

In the current manuscript, the authors identified MIK2 as essential for SCOOP/STMP peptide-induced responses, likely by direct binding, and showed data that BAK1/BKK1 and BIK1/PBL1 are important for responses to SCOOPs. Notably, they also showed that some species in *Fusarium* and *Comamonadaceae* contain a conserved sequence similar to SCOOPs that can also induce defense responses in a MIK2-dependent manner. The authors suggest that these SCOOPs/STMPs are endogenous regulators of defense induction akin to DAMPs and may have evolved to mimic microbial SCOOP-like proteins (SCOOPs). All data presented were convincing. I have several comments/questions below to help the function of these immunogenic peptides and their receptors substantially clearer.

1. Recent papers (<https://doi.org/10.1093/jxb/ery454> and <https://doi.org/10.1111/jipb.12817>) identified SCOOPs/STMPs in *Arabidopsis*, and noted that these proteins appear to be restricted to the Brassicaceae family. The authors need to specify this fact in the abstract (and also at the last sentence on page 12) that helps the audiences focus on an interesting discussion point how this immunogenic peptides have evolved in certain species in plants, fungi, and bacteria for plant immunity.

2. Are SCOOP pro-proteins processed in planta to produce the small peptides as used in the experiments? This question is not addressed experimentally nor discussed in this manuscript. The second to last sentence in the discussion, "...plant SCOOPs are generated from peptide precursor proteins...", is not supported by the data, nor from the above two papers on SCOOPs/STMPs, as far as I can tell.

3. Similarly, what is the subcellular localization of these proteins: are they in apoplastic fluid or associated with membranes? The STMP paper in JIPB suggested that some of these SCOOPs, including some tested here and found to be active, i.e., SCOOP4, 15, 13, and 14 are transmembrane proteins, and may be proteolytically processed. Where are they really? This question and the question 2 could be possibly be addressed by mass spectrometry experiments.

4. In the same vein, are the putative SCOOP-like protein sequences found in the *Fusarium* species processed in planta to produce small active peptides?

5. Are *Fusarium* mutants lacking the identified putative SCOOP-like protein differently virulent? It would be nice to show such data in combination with an experiment shown in the previous paper in *PLoS Path* (<https://doi.org/10.1371/journal.pgen.1006832>) in which *mik2* mutants were found to be more susceptible to *Fusarium oxysporum*.

Minor comments

What is “physiological responses” in Abstract. Please specify.

I disagree with the statement “DAMPs are usually induced upon MAMP perception” given that there are other type of DAMPs, e.g., ATP, OGs, and HMGBs. The authors probably meant “peptidic DAMPs” in the context of Introduction.

Please add proper descriptions in the figure legends. For example, some descriptions for Figure 2-5 were halfway done or missing. I could not evaluate the data properly.

The lower panel of figure 5h should include "- SCOOP12" to contrast with the upper panel "+SCOOP12"

The bands in Figure 5h and 5j were indicated wrongly by MIK2 and BAK1. It should be a complex of MIK2-BAK1 and a mixture of individual MIK2 and BAK1, respectively.

Why SCOOP10 still inhibit the root growth partially in bik1/pbl1 in Figure 5i and 5m? Explain in the text.

The root pictures are too blurry in Supplementary Figure 1B.

The last part of Result left a few questions. There is no explanation why At SCOOP1 lost inhibitory effect on the root growth in comparison to Cu and Ve SCOOP1s. Do other SCOOP1s (from As to Asn) have any activities as a DAMP? I am curious about Asj and Asn in comparison to At whether insertion of G7 and G/S8 can overcome the lost activity of At due to P4.

GUS expression seems to be inhibited after treated with At and Aa SCOOP1 in Supplementary Figure 7. If these are consistent data, add a note in the text.

The author discussed that no enhanced susceptibility to Fusarium in the scoop12 mutant could be due to the functional redundancy of SCOOP1s (page 13). But it could be due to Fusarium SCOOP1s as well.

Reviewer #3 (Remarks to the Author):

In the manuscript entitled “Immune elicitation by sensing the conserved signature from phytochemicals and microbes via the Arabidopsis MIK2 receptor”, the authors present evidence that SCOOP peptides define ligands for MIK2 receptor in MIK2-regulated immune and physiological responses. Characterization of plants expressing chimeric RLK7/PEPR1-MIK2 receptors suggest that MIK2 kinase domain (KD) can specifically induce MIK2-mediated responses. A chimeric receptor analysis suggested both common and distinctive aspects of MIK2 signaling compared to RLK7 and PEPR1 signaling. SCOOP genes were among the MIK2-specific, PIP1-induced genes in RLK7-MIK2 plants. Genetic and biochemical studies indicate that SCOOP peptides act as MIK2 ligands to activate PTI-like immune responses. Moreover, the authors describe elicitor-active SCOOP-like peptides in *Fusarium* fungi, whose application induces PTI-like defense responses in *Arabidopsis*, in a manner dependent on MIK2. The paper (including methods) is well written, and makes significant advance into plant cell signaling and immunity regulation via peptide ligands and receptor kinases. In conclusion, this work is valuable and can appeal to the readership of the journal, but it needs to be further elaborated in the following aspects.

Major concerns:

1) The data suggest that MIK2TK has distinctive properties compared to RLK7 or PEPR1 KDs, when fused to the ECD of the latter RKs. It is also possible, however, that defense hyper-activation by RLK1/PEPR1-MIK2 merely results from an increased MIK2TK dose. Whether the observed effects reflect the inherent function of MIK2, in the full-length protein context, remains to be shown. If not, the authors need cautions in the interpretation of the data.

Is MIK2TK alone or MIK2 overexpression (but not the ED-TK chimera) sufficient to trigger enhanced PTI-like responses and mimic SCOOP application?

2) Following the significant finding that MIK2 identifies a SCOOP receptor, the authors are expected to address how much of the previously described MIK2 functions can be explained by SCOOPs and vice versa, including pollen responses to LURE, cell wall disintegration responses and salt stress tolerance? SCOOPs are specifically seen in Brassicaceae, but how about MIK2? Careful comparisons and discussions between SCOOP and MIK2 functions, expression and distribution are appreciated. Gully et al 2019 present a microarray profile between wild type and *proscop12* mutant. A comparison between MIK2TK- and PROSCOOP12-dependent genes should give some insight.

In addition, the molecular basis and the significance remain to be solved for the observed MIK2 TK-specific browning. How does this relate to immunity or root growth regulation? Although this may be beyond the scope of this paper, it should be precisely stated that browning sites are different between MIK2TK-induced and SCOOP-induced root browning (Suppl Fig 2 and Fig 2).

3) It is striking to show that fungal SCOOP sequences are also elicitor-active. Consistent with the authors' idea that these SCOOPs provide MAMPs, it was previously shown that *Fusarium* resistance is reduced in *mik2* mutants. However, since *mik2* is insensitive to both fungal and plant SCOOPS, it is not clear whether fungal hyper-susceptibility of *mik2* attributes to the loss of fungal SCOOP recognition. Fungal pathogens globally and dynamically reprogram gene expression during plant infection. To strengthen the significance of the findings, it is important and in my view feasible to show that these fungal proteins (or minimally genes) are expressed during their infection in plants.

In this respect, Gully et al 2019 showed that *proscop12* mutants exhibit enhanced resistance against necrogenic bacterium *Erwinia amylovora* and necrotrophic fungus *Alternaria brassicola*. On the other hand, SCOOP12 pretreatment induces resistance against hemibiotrophic bacterium *Pseudomonas syringae*. A comprehensive model for SCOOPs and MIK2 in plant immunity needs to be provided in light of the previous and present studies, favorably with a model figure.

Detailed points:

4) In Fig 2, it should be mentioned that the extents of root browning, growth inhibition and defense activation are not correlated, among the tested SCOOPs.

5) Kind explanation is appreciated in Fig 2g and 2m. Roots look so different following SCOOP application, which may confuse non-expert readers not knowing the differences in the assay conditions.

6) The data may be repetitively shown in Fig 2e and 3c for WT.

7) When testing *mik2* mutants, why did you use SCOOP10 for root growth inhibition and SCOOP12 for defense responses in Fig 3?

8) I agree that TAMRA-SCOOP12 is bioactive, but seems less active compared to SCOOP12. The results need to be precisely described.

9) It is not clear which root regions were stained with TAMRA-SCOOP12 and shown in Fig 4. Does this reflect the expression sites for SCOOP12 gene expression sites?

10) According to the calculated K_d 1.78 μM in Fig 4f, the authors suggest high affinity between MIK2ECD and SCOOP12. What is the basis for their argument? In comparison with other RK-peptide ligand pairs? Or, is there any data for physiological concentrations of SCOOPs in plants? In this respect, it should also be noted that SCOOP10B responses were detectable in planta in a nM range (Fig 2e).

11) BAK1 and BIK1/PBL1 requirements were previously described in SCOOP12- and MIK2-mediated defense responses (Coleman, *New Phytologist*, 2020; Gully, *J Exp Bot* 2019). Consistency with previous studies needs to be mentioned and discussed.

12) On page 10, the SCOOP-MIK2-BAK1 receptorsome participates conserved signaling pathways shared by other LRR-containing PRR complexes. It cannot be generalized to MAMP/DAMP/PRRs.

13) Page 13, extracellular release of EF-Tu has been described in previous studies, which need to be cited and discussed. Some studies can be found in e.g. Harvery et al, *Front Microbiol* 2019 (doi.org/10.3389/fmicb.2019.02351) and He et al, *Front Microbiol* 2015 (doi.org/10.3389/fmicb.2015.00707).

14) A phylogenetic analysis for representative members of plant and microbial SCOOP peptides is helpful and appreciated to deepen the discussions on their evolutionary aspects.

15) The mechanisms by which the plant PROSCOOPs or SCOOP peptides are released is not shown, and even whether this question is solved or unsolved is not mentioned in the text. Do they have an N-terminal secretion signal?

We thank the editor and reviewers for their thorough reviews of our manuscript and insightful comments, to which we respond point-by-point below (in blue). The following are several key experiments we performed to address reviewers' concerns and strengthen our conclusions:

- 1) We have detected SCOOP12-GFP localization and expression in the *pSCOOP12::SCOOP12-GFP/scoop12* transgenic plants (Supplementary Figures 3c, d). The data suggest that SCOOP12-GFP is likely proteolytically processed and secreted to apoplasts. We also provided a schematic diagram showing SCOOP domains and the potential processing from the precursor proteins (Supplementary Figures 2b, 3b).
- 2) We have performed RNA-Seq analysis to show that SCOOP12-regulated genes largely overlap with genes activated in *RLK7^{ECD}-MIK2^{TK}/rlk7* transgenic plants treated with PIP1 (Supplementary Figures 4f-h).
- 3) We detected the expression of gene carrying *SCOOP-LIKE* (*SCOOP-LIKE*) sequence in *Fusarium* during its infection on *Arabidopsis* (Supplementary Figure 7d). We further knocked-out *SCOOP-LIKE* in *F. oxysporum* strain *Fo5176* and the mutants showed the elevated virulence on *Arabidopsis* (Supplementary Figures 8a-d). Consistently, the *mik2* mutant was more susceptible to *Fo5176* (Supplementary Figures 8e, f);
- 4) We performed multiple phylogenetic analyses of MIK2, SCOOPs in *Arabidopsis* and microbes and discussed their evolution (Supplementary Figures 2b, 4d, 7c, 9a, and 10b).
- 5) We provided a model for SCOOP/SCOOP-LIKE-MIK2 roles in plant immunity (Supplementary Figure 10a).

Reviewer #1 (Remarks to the Author):

Hou et al. reported that SCOOP peptides trigger diverse immune responses via leucine rich-repeat receptor kinase MIK2. They first observed that, when the chimeric RLK7(ECD)-MIK2(TK) receptor was overexpressed in the *rlk7* mutant, SCOOP and STMP peptides were induced after PIP1 treatment. Then they analyzed SCOOP peptides and MIK2 receptor in detail and proposed that SCOOP and MIK2 function as ligand-receptor pair involved in immune responses. The *mik2-1* and *mik2-2* mutants were insensitive to SCOOP peptides, and SCOOP peptide directly binds extracellular LRR domain of MIK2 with a dissociation constant (Kd) of approximately 1-3 μ M. The authors also showed that BAK1 and SERK4 are co-receptors for MIK2 in mediating SCOOP-triggered immunity. In addition, the MIK2-BAK1 receptor complex activation by SCOOPs triggers phosphorylation of receptor-like cytoplasmic kinases BIK1 and PBL1. Interestingly, SCOOP-like motifs are highly conserved in various microbes, and they may serve as MAMPs perceived by the plant MIK2 receptor.

We thank the reviewer for the concise summary and insightful comments on our work.

The data was clearly presented, the manuscript was well written, and the take-home message of the work is straightforward. Here are some comments.

(1) The authors reported that, when the chimeric RLK7(ECD)-MIK2(TK) receptor was overexpressed in the *rlk7* mutant, SCOOP and STMP peptides were induced after PIP1 treatment. This should be important data that inspired the authors to focus on SCOOP peptides, but there is no explanation for the rank of SCOOP genes among all differently expressed genes. Many readers will be surprised at the sudden emergence of the SCOOP genes at this section.

Our response: We thank the reviewer for the constructive comment. As suggested, we have added the new Supplementary Figure 2a, and more information in the Results (Page 5, Line 18-23). We analyzed RNA-Seq data in PIP1-treated *RLK7^{ECD}-MIK2^{TK}/rlk7*

transgenic plants and identified 51 candidate genes encoding small secreted peptides among PIP1-upregulated genes. Eighteen genes out of 51 (>35%) encode SCOOPs, STMP and SCOOP/STMP-like genes, which prompted us to investigate SCOOPs and STMPs in the following studies.

(2) SCOOP10B is active even at 1 nM but binding constant between SCOOP and MIK2 is approximately 1-3 μ M. The authors need to elaborate on this discrepancy.

Our response: Thanks for pointing this out. This might be due to the differences between *in vivo* and *in vitro* assays. First, as shared coreceptors of many LRR-RKs, including MIK2 as shown here, BAK1 and SERKs might promote ligand-receptor binding *in vivo*. For example, IDA-HAESA interaction displayed a dissociation constant of 20 μ M without SERKs. It's approximately 60 times weaker than the affinity between IDA and HAESA in presence of SERK1 ($K_d = 350$ nM) (Santiago et al., 2016). Second, interactions between transmembrane or cytoplasmic domains within the cellular context might provide an environment more favorable for the SCOOP-MIK2 interaction. For example, microsomal fractions derived from cells showed that PSK-PSKR interaction displayed a dissociation constant of 4.2 nM in carrot and 7.7 nM in *Arabidopsis*, approximately 370 times higher than the affinity measured between PSK^{LRR} and PSK with *in vitro* microscale thermophoresis (MST) analysis (Wang et al., 2015). Third, other unknown components might also stimulate SCOOP-MIK2 binding *in vivo*. Notably, this is a common observation in the studies of ligand-receptor interactions. For instance, flg22 is highly active at nM scales *in vivo*, but the K_d of flg22-FLS2 binding is at μ M scale (Sun et al., 2013, Okuda et al., 2020).

(3) Is there any information about the *in vivo* natural structure of SCOOP peptides? All experiments are performed on synthetic peptides based on the predicted putative structure.

Our response: SCOOP peptides are likely derived from SCOOP precursor proteins, which harbor an N-terminal signal peptide for directing peptide secretion, variable regions and C-terminal functional peptide region. The precursor proteins are usually processed to release the functional peptides (Olsson et al., 2019). We have provided a schematic diagram showing SCOOP domains (new Supplementary Figures 2b, 3b). We also predicted the structure of SCOOP12 using I-TASSER (new Supplementary Figure 3a). In addition, we detected SCOOP expression at the plasma membrane and intercellular spaces in *pSCOOP12::SCOOP12-GFP* transgenic plants (new supplementary Figure 3c), suggesting that SCOOP12 is likely secreted. Further, SCOOP12 might be processed in plants as indicated by additional bands in immunoblots (new supplementary Figure 3d). These data were described in Results (Page 6, Line 13-25).

(4) Is *mik2* mutant vulnerable to infection with fungal *Fusarium* spp. or bacterial Comamonadaceae?

Our response: Thanks for bringing this up. We have included new data to show that the *mik2* mutant is more susceptible to *Fusarium oxysporum* Fo5176 than wild-type Col-0 plants (new supplementary Figures 8e, f). This result is consistent with a previous report (Dieuwertje Van der Does et al., 2017).

Reviewer #2 (Remarks to the Author):

In the current manuscript, the authors identified MIK2 as essential for SCOOP/STMP peptide-induced responses, likely by direct binding, and showed data that BAK1/BKK1 and BIK1/PBL1 are important for responses to SCOOPs. Notably, they also showed that some species in *Fusarium* and Comamonadaceae contain a conserved sequence similar to SCOOPs that can also induce defense responses in a MIK2-dependent manner. The authors suggest that these SCOOPs/STMPs are endogenous regulators of defense

induction akin to DAMPs and may have evolved to mimic microbial SCOOP-like proteins (SCOOPs). All data presented were convincing. I have several comments/questions below to help the function of these immunogenic peptides and their receptors substantially clearer.

We thank the reviewer for the comments and the recognition of our work.

1. Recent papers (<https://doi.org/10.1093/jxb/ery454> and <https://doi.org/10.1111/jipb.12817>) identified SCOOPs/STMPs in *Arabidopsis*, and noted that these proteins appear to be restricted to the Brassicaceae family. The authors need to specify this fact in the abstract (and also at the last sentence on page 12) that helps the audiences focus on an interesting discussion point how this immunogenic peptides have evolved in certain species in plants, fungi, and bacteria for plant immunity.

Our response: We thank the reviewer for the suggestion. We have added this information in the Abstract (Page 2, Line 6), Discussion (Page 13, Line 18) and further discussed their evolution in the last paragraph of Discussion.

2. Are SCOOP pro-proteins processed in planta to produce the small peptides as used in the experiments? This question is not addressed experimentally nor discussed in this manuscript. The second to last sentence in the discussion, "...plant SCOOPs are generated from peptide precursor proteins...", is not supported by the data, nor from the above two papers on SCOOPs/STMPs, as far as I can tell.

Our response: We thank the reviewer for the suggestion. We have included new data showing that SCOOP12-GFP could be detected at the plasma membrane and extracellular spaces in *Arabidopsis pSCOOP12::SCOOP12-GFP/scoop12* transgenic plants (new Supplementary Figure 3c), suggesting that SCOOP12 is likely secreted. Further, SCOOP12 might be processed in plants as indicated by additional bands in immunoblots (new Supplementary Figure 3d). These data were described in Results (Page 6, Line 13-25).

3. Similarly, what is the subcellular localization of these proteins: are they in apoplastic fluid or associated with membranes? The STMP paper in JIPB suggested that some of these SCOOPs, including some tested here and found to be active, i.e., SCOOP4, 15, 13, and 14 are transmembrane proteins, and may be proteolytically processed. Where are they really? This question and the question 2 could be possibly be addressed by mass spectrometry experiments.

Our response: We thank the reviewer for the suggestions. As mentioned above, we have performed SCOOP12-GFP localization studies using the *pSCOOP12::SCOOP12-GFP/scoop12* transgenic plants in which the *scoop12* mutant was transformed with *SCOOP12-GFP* under the control of its native promoter. We observed that SCOOP12-GFP is associated with the plasma membrane in transgenic plants. After plasmolysis, SCOOP12-GFP was also detected in apoplasts, suggesting that SCOOP12-GFP was likely proteolytically processed and secreted to apoplasts. The new data are in Supplementary Figure 3c, and the results were described in Page 6, Line 13-25.

4. In the same vein, are the putative SCOOP-like protein sequences found in the *Fusarium* species processed in planta to produce small active peptides?

Our response: *Fusarium* SCOOP-LIKE motif is located in the N-terminus of a highly conserved transcription factor. In the revised manuscript, we provided new data to show that this gene was induced upon *Fusarium* infections in *Arabidopsis* (new Supplementary Figure 7d). We agree with this reviewer that it will be interesting to follow up whether and how the peptides are processed *in planta*. Notably, elf18, a well-studied MAMP, is derived from the conserved and cytoplasm-localized bacterial translation elongation factor EF-Tu

(Kunze et al., 2004). Recent studies show that EF-Tu could be detected on the extracellular surface and secretome of bacteria (Harvey et al., 2019; He et al., 2015). We have discussed this in the second paragraph of Page 15.

5. Are *Fusarium* mutants lacking the identified putative SCOOP-like protein differently virulent? It would be nice to show such data in combination with an experiment shown in the previous paper in PLoS Path (<https://doi.org/10.1371/journal.pgen.1006832>) in which *mik2* mutants were found to be more susceptible to *Fusarium oxysporum*.

Our response: We thank this reviewer's suggestion. We have been working on this since we identified SCOOP-LIKE sequence in *Fusarium*. We generated two mutant lines of *Fusarium oxysporum* Fo5176 lacking SCOOP-LIKE gene by homologous recombination. Compared to the WT *Fusarium* strain, *Fusarium scoop-like* mutants showed elevated virulence on *Arabidopsis* supporting the notion that *Fusarium* SCOOPs function as a MAMP (new Supplementary Figures 8a-d). As suggested, we also performed *Fusarium* infection assays in the *mik2* mutant. Consistent with the previous report (Dieuwertje Van der Does et al., 2017), the *mik2* mutant was more susceptible to *Fusarium* infections than WT plants (new Supplementary Figures 8e, f). These new results were described in Page 12, Line 18-23.

Minor comments

1. What is "physiological responses" in Abstract. Please specify.

Our response: We have specified "physiological responses" to "altered root development".

2. I disagree with the statement "DAMPs are usually induced upon MAMP perception" given that there are other type of DAMPs, e.g., ATP, OGs, and HMGBs. The authors probably meant "peptidic DAMPs" in the context of Introduction.

Our response: Thanks for the suggestion. We changed it to "peptidic DAMPs" as suggested.

3. Please add proper descriptions in the figure legends. For example, some descriptions for Figure 2-5 were halfway done or missing. I could not evaluate the data properly.

Our response: We have added more detailed descriptions in the figure legends as suggested.

4. The lower panel of figure 5h should include "-SCOOP12" to contrast with the upper panel "+SCOOP12"

Our response: We have added "- SCOOP12" in the lower panel of figure 5h.

5. The bands in Figure 5h and 5j were indicated wrongly by MIK2 and BAK1. It should be a complex of MIK2-BAK1 and a mixture of individual MIK2 and BAK1, respectively.

Our response: In Figures 5h and 5j, SDS-PAGE was used for the gel filtration assay of MIK2-BAK1 complex formation. The protein complex would be mostly depolymerized by SDS in the loading buffer and SDS-PAGE. Thus, the top band corresponds to MIK2^{ECD} and the bottom band corresponds to BAK1^{ECD}. The similar assays and presentations were also shown in publications for other receptor-coreceptor complexes, including FLS2-BAK1 (Sun et al., 2013, Figure S1B), PEPR1-BAK1 (Tang et al., 2014, Figure 1B), and PSKR1-SERK1 (Wang et al., 2015, Figure 2a and Extended Data Figure 4 & 8).

6. Why SCOOP10 still inhibit the root growth partially in *bik1/pbl1* in Figure 5i and 5m?

Explain in the text.

Our response: RLCKs have multiple members. It is possible that additional members of RLCKs may function redundantly with BIK1 and PBL1 downstream of SCOOP-MIK2. Alternatively, SCOOP-MIK2 may also activate a RLCK-independent pathway. We have discussed these possibilities in our revised manuscript (Page 11, Line 4-6).

7. The root pictures are too blurry in Supplementary Figure 1B.

Our response: We apologize for this. We have repeated the experiment and got the similar results. We have zoomed-in the images for a clear phenotype observation, and labeled the hypocotyl-root junction region with a red arrow.

8. The last part of Result left a few questions. There is no explanation why At SCOOPPL lost inhibitory effect on the root growth in comparison to Cu and Ve SCOOPPLs. Do other SCOOPPLs (from As to Asn) have any activities as a DAMP? I am curious about Asj and Asn in comparison to At whether insertion of G7 and G/S8 can overcome the lost activity of At due to P4.

Our response: We have included the explanation in which sequence variations in the fourth residue in AtSCOOPPL and in the first conserved serine residue in AaSCOOPPL may contribute the inactivity of AtSCOOPPL and AaSCOOPPL compared to CuSCOOPPL and VeSCOOPPL (Supplementary Figure 9a) (Page 12, Line 30-31). We have also extended the phylogenetic analysis to include all *Comamonadaceae* bacteria with SCOOPPL sequences in new Supplementary Figure 9a. We agree with this reviewer that it will be interesting to investigate whether all other SCOOPPLs from bacteria have any activities as a MAMP. Here, we intend to provide a proof-of-concept that some microbial SCOOPPLs, including ones from fungi and bacteria, could activate MIK2-dependent immune responses. Considering a large number of SCOOPPLs in different *Comamonadaceae* bacteria and scope-of-work of our current manuscript (identification of MIK2 as the receptor for plant SCOOPs and microbial SCOOPPLs in plant immunity), we hope that this reviewer would agree that such efforts could be a future following-up project.

9. GUS expression seems to be inhibited after treated with At and Aa SCOOPPL in Supplementary Figure 7. If these are consistent data, add a note in the text.

Our response: We apologize for having misled this reviewer. Comparing data from multiple repeats of this experiment, we did not observe a claimable inhibitory effect of At and AaSCOOPPLs on GUS expression. We have replaced a representative image in Supplementary Figure 9d.

10. The author discussed that no enhanced susceptibility to *Fusarium* in the scoop12 mutant could be due to the functional redundancy of SCOOPs (page 13). But it could be due to *Fusarium* SCOOPPLs as well.

Our response: Thanks for the suggestion. We have included this possibility in the discussion (Page 15, Line 2-3).

Reviewer #3 (Remarks to the Author):

In the manuscript entitled “Immune elicitation by sensing the conserved signature from phyto cytokines and microbes via the Arabidopsis MIK2 receptor“, the authors present evidence that SCOOP peptides define ligands for MIK2 receptor in MIK2-regulated immune and physiological responses. Characterization of plants expressing chimeric RLK7/PEPR1-MIK2 receptors suggest that MIK2 kinase domain (KD) can specifically induce MIK2-mediated responses. A chimeric receptor analysis suggested both common

and distinctive aspects of MIK2 signaling compared to RLK7 and PEPR1 signaling. SCOOP genes were among the MIK2-specific, PIP1-induced genes in RLK7-MIK2 plants. Genetic and biochemical studies indicate that SCOOP peptides act as MIK2 ligands to activate PTI-like immune responses. Moreover, the authors describe elicitor-active SCOOP-like peptides in *Fusarium* fungi, whose application induces PTI-like defense responses in *Arabidopsis*, in a manner dependent on MIK2. The paper (including methods) is well written, and makes significant advance into plant cell signaling and immunity regulation via peptide ligands and receptor kinases. In conclusion, this work is valuable and can appeal to the readership of the journal, but it needs to be further elaborated in the following aspects.

We thank the reviewer for the comments and the recognition of our work.

Major concerns:

1) The data suggest that MIK2TK has distinctive properties compared to RLK7 or PEPR1 KDs, when fused to the ECD of the latter RKs. It is also possible, however, that defense hyper-activation by RLK1/PEPR1-MIK2 merely results from an increased MIK2TK dose. Whether the observed effects reflect the inherent function of MIK2, in the full-length protein context, remains to be shown. If not, the authors need cautions in the interpretation of the data.

Is MIK2TK alone or MIK2 overexpression (but not the ED-TK chimera) sufficient to trigger enhanced PTI-like responses and mimic SCOOP application?

Our response: We thank the reviewer for the suggestion. We have included the discussion of this possibility in the revised manuscript (Page 5, Line 15-17). Notably, the observed MIK2 effects were apparent only upon ligand treatments. The *RLK7^{ECD}-MIK2^{TK}/rlk7* transgenic plants showed similar phenotypes as WT or *RLK7/rlk7* transgenic plants in the absence of elicitor treatment (Figures 1b, c). We have included the data of H₂O treatment for ROS burst in the revised Figure 1d. In addition, RNA-Seq analysis indicated that compared to water treatment control, PIP1 treatment upregulated a large number of genes in *RLK7^{ECD}-MIK2^{TK}/rlk7* transgenic plants, which largely overlapped with and were clustered together SCOOP12-upregulated genes (Supplementary Fig. 3f, g). Thus, it is likely that MIK2TK alone may not be sufficient to trigger enhanced PTI-like responses.

2) Following the significant finding that MIK2 identifies a SCOOP receptor, the authors are expected to address how much of the previously described MIK2 functions can be explained by SCOOPs and vice versa, including pollen responses to LURE, cell wall disintegration responses and salt stress tolerance?

Our response: We thank the reviewer for the suggestion. As noted, MIK2 has been implicated to play a role in multiple biological processes, including salt stress, cell wall integrity sensing and plant resistance to *Fusarium*. Our primary focus of this manuscript is to identify MIK2 as the receptor of SCOOPs in plant immunity. As this and other reviewers suggested, we have added the new data to show that the *mik2* mutant is more susceptible to *Fusarium* infections (new Supplementary Figures 8e, f). We did not observe the enhanced susceptibility to *F. oxysporum* in the *scoop12* mutant compared to WT plants, likely due to the functional redundancy of different SCOOPs in *Arabidopsis*. Considering that a large number of SCOOPs trigger MIK2-dependent immune responses, higher-order *scoop* mutants are likely needed to dissect the diverse phenotypes observed in the *mik2* mutant. We are in the process of generating a series of higher-order *scoop* mutants by multiple genetic crosses of different combinations of T-DNA insertion mutants and generating CRISPR-Cas9 gene editing lines. We agree with this reviewer that it is interesting to systematically dissect the contributions of different SCOOPs and MIK2 in diverse biological processes, including plant growth and development, and abiotic and biotic stress responses when we obtain these valuable genetic resources. We hope that this reviewer would agree that this could be a future follow-up project in light of the main

focus of our manuscript on the identification of MIK2 as the receptor of SCOOP peptides in plant immunity.

SCOOPs are specifically seen in Brassicaceae, but how about MIK2?

Our response: Phylogenetic analysis indicated that similar to SCOOPs, the apparent MIK2 orthologs with > 60% amino acid identity to *Arabidopsis* MIK2 were identified only in Brassicaceae plants (new Supplementary Fig. 4d).

Careful comparisons and discussions between SCOOP and MIK2 functions, expression and distribution are appreciated.

Our response: We thank the reviewer for the suggestion. We have provided RT-qPCR data to show tissue-specific expression of different SCOOPs in *Arabidopsis* (Figure 2b). We also performed RT-qPCR analysis of *MIK2* expression. The data show that *MIK2* is ubiquitously expressed in shoots, roots, and leaves (new Supplementary Fig. 4e), which is consistent with eGFP information (<http://bar.utoronto.ca/efp/cgi-bin/efpWeb.cgi>). It is possible that MIK2 regulates diverse physiological responses in different tissues or cell types upon the perception of different members of SCOOPs. We have included this in the discussion (Page 14, Line 24-28).

Gully et al 2019 present a microarray profile between wild type and proscop12 mutant. A comparison between MIK2TK- and PROSCOOP12-dependent genes should give some insight.

Our response: We thank the reviewer for the suggestion. We performed RNA-Seq analyses and compared genes regulated by SCOOP12 treatment in WT with genes regulated by MIK2 kinase domain in *RLK7^{ECD}-MIK2^{TK}/rlk7* transgenic plants upon PIP1 treatment. The venn diagram and heat map analysis indicated that SCOOP12-regulated genes largely overlapped and were clustered together with MIK2 kinase domain-regulated genes in *RLK7^{ECD}-MIK2^{TK}/rlk7* plants upon PIP1 treatment compared to PIP1-regulated genes in WT (Supplementary Figures 4f, g). Similar to PIP1 treatment in *RLK7^{ECD}-MIK2^{TK}/rlk7* transgenic plants, SCOOP12 treatment induced the expression of different SCOOPs (Supplementary Fig. 4h). The data further support largely overlapping responses between SCOOP12 perception and MIK2 activation. The results were presented in the last paragraph in Page 8.

In addition, the molecular basis and the significance remain to be solved for the observed MIK2TK-specific browning. How does this relate to immunity or root growth regulation? Although this may be beyond the scope of this paper, it should be precisely stated that browning sites are different between MIK2TK-induced and SCOOP-induced root browning (Suppl Fig 2 and Fig 2).

Our response: We agree with this reviewer that the molecular basis and the significance of the observed root browning are not clear at the moment. It is possible that the browning might be related to ROS production or cell wall modifications, such as lignification and suberification, upon SCOOP-MIK2 activation. Different browning sites between MIK2^{TK}-induced and SCOOP-induced root browning might be associated with the potential *MIK2* expression differences in *35S::RLK7^{ECD}-MIK2^{TK}* transgenic plants and WT plants. We have included these discussions in the revised manuscript (Page 7, Line 4-8).

3) It is striking to show that fungal SCOOP sequences are also elicitor-active. Consistent with the authors' idea that these SCOOPs provide MAMPs, it was previously shown that *Fusarium* resistance is reduced in *mik2* mutants. However, since *mik2* is insensitive to both fungal and plant SCOOPs, it is not clear whether fungal hyper-susceptibility of *mik2* attributes to the loss of fungal SCOOP recognition.

Our response: Thanks for this comment. We have generated two lines of *Fusarium oxysporum* 5176 mutants lacking *SCOOP-LIKE* peptide sequence by homologous recombination. Our data showed that *Fusarium scoop-like* mutants showed elevated virulence on *Arabidopsis* compared to WT strain (Supplementary Figures 8a-d). The data suggest that recognition of fungal SCOOPL contributes to plant resistance to *Fusarium* infections, supporting that fungal SCOOPL functions as a MAMP.

Fungal pathogens globally and dynamically reprogram gene expression during plant infection. To strengthen the significance of the findings, it is important and, in my view, feasible to show that these fungal proteins (or minimally genes) are expressed during their infection in plants.

Our response: Thanks for the suggestion. We have performed RT-qPCR analysis and detected the expression of gene carrying *SCOOP-LIKE* sequence in *Fusarium oxysporum* Fo5176 during infections in *Arabidopsis*. The data show that the gene (*FOXG_11846*) harboring *SCOOPL* sequence was expressed, and the expression was upregulated at 48 hours upon fungal inoculation (new Supplementary Fig. 7d).

In this respect, Gully et al 2019 showed that *proscop12* mutants exhibit enhanced resistance against necrogenic bacterium *Erwinia amylovora* and necrotrophic fungus *Alternaria brassicicola*. On the other hand, *SCOOP12* pretreatment induces resistance against hemibiotrophic bacterium *Pseudomonas syringae*. A comprehensive model for SCOOPs and MIK2 in plant immunity needs to be provided in light of the previous and present studies, favorably with a model figure.

Our response: We have provided a model figure to show that MIK2 perceives plant endogenous SCOOPs and SCOOPs from microbes to activate plant immunity. We also integrated previous studies about the resistance against necrogenic bacterium *Erwinia amylovora*, necrotrophic fungus *Alternaria brassicicola*, and hemibiotrophic bacterium *Pseudomonas syringae* in the model (new Supplementary Fig. 10a). The model also includes MIK2 complexing with BAK1 and SERK4.

Detailed points:

4) In Fig 2, it should be mentioned that the extents of root browning, growth inhibition and defense activation are not correlated, among the tested SCOOPs.

Our response: Thanks for the suggestion. We have noted this in our revised manuscript (Page 7, Line 12-13).

5) Kind explanation is appreciated in Fig 2g and 2m. Roots look so different following SCOOP application, which may confuse non-expert readers not knowing the differences in the assay conditions.

Our response: Thanks for the suggestion. We described the Figure 2m (Figure 2l in the revised manuscript) in more details. We have included a sentence to explain that "Arabidopsis root tips with SCOOP peptide treatments for 3 hr have no different developmental phenotypes from H₂O treatment." (Page 29, Line 1-2).

6) The data may be repetitively shown in Fig 2e and 3c for WT.

Our response: Thanks for pointing this out. Yes, WT was shared in the original Figures 2e and 3c. Since Figure 3c included the information in the original Figure 2e, we deleted Figure 2e in the revised manuscript.

7) When testing *mik2* mutants, why did you use SCOOP10 for root growth inhibition and SCOOP12 for defense responses in Fig 3?

Our response: SCOOP10 and SCOOP12 behaved similarly for root growth inhibition and defense responses. In the revised manuscript, we have included SCOOP10 for ROS burst and MAPK activation assays (Figures 3f, g).

8) I agree that TAMRA-SCOOP12 is bioactive, but seems less active compared to SCOOP12. The results need to be precisely described.

Our response: We have noted that TAMRA-SCOOP12 is slightly less active than SCOOP12 (Page 9, Line 5-6).

9) It is not clear which root regions were stained with TAMRA-SCOOP12 and shown in Fig 4. Does this reflect the expression sites for SCOOP12 gene expression sites?

Our response: Root tips were stained with TAMRA-SCOOP12. Our RT-qPCR analysis showed that *SCOOP12* expressed highly in roots (Figure 2b).

10) According to the calculated K_d 1.78 μM in Fig 4f, the authors suggest high affinity between MIK2^{ECD} and SCOOP12. What is the basis for their argument? In comparison with other RK-peptide ligand pairs? Or, is there any data for physiological concentrations of SCOOPs in plants? In this respect, it should also be noted that SCOOP10B responses were detectable in planta in a nM range (Fig 2e).

Our response: We apologize for the confusion. The calculated K_d of 1.78 μM for MIK2^{ECD}-SCOOP12 is comparable to 1.5 μM of FLS2-flg22 (Okuda et al., 2019, Fig. 1c). To avoid confusion, we have deleted “suggesting a high affinity of MIK2^{ECD} with SCOOP12” in the revised manuscript. We have explained above in response to Reviewer 1, Point 2, in terms of possibilities why SCOOP10^B was active at an nM scale *in planta* while at an μM scale in the *in vitro* binding assays. This might be due to the differences between *in vivo* and *in vitro* assays. First, as shared coreceptors of MIK2, BAK1 and SERKs could promote ligand-receptor binding *in vivo*. Second, interactions between transmembrane or cytoplasmic domains within the cellular context might provide an environment more favorable for the SCOOP-MIK2 interaction. Third, other unknown components may also stimulate SCOOP-MIK2 binding *in vivo*. Notably, flg22 is highly active at a nM scale *in vivo*, but the K_d of flg22-FLS2 binding is at a μM scale (Sun et al., 2013; Okuda et al., 2019).

11) BAK1 and BIK1/PBL1 requirements were previously described in SCOOP12- and MIK2-mediated defense responses (Coleman, New Phytologist, 2020; Gully, J Exp Bot 2019). Consistency with previous studies needs to be mentioned and discussed.

Our response: Thanks for pointing this out. We have cited these two papers and discussed along with our results (Page 10, Line 3; Page11, Line 10-11).

12) On page 10, the SCOOP-MIK2-BAK1 receptorsome participates conserved signaling pathways shared by other LRR-containing PRR complexes. It cannot be generalized to MAMP/DAMP/PRRs.

Our response: Thanks for the suggestion. We have revised it as suggested.

13) Page 13, extracellular release of EF-Tu has been described in previous studies, which need to be cited and discussed. Some studies can be found in e.g. Harvery et al, Front Microbiol 2019 (doi.org/10.3389/fmicb.2019.02351) and He et al, Front Microbiol 2015 (doi.org/10.3389/fmicb.2015.00707).

Our response: Thanks for the suggestion. We have cited and discussed these two references in the Discussion (Page 15, Line 14-15).

14) A phylogenetic analysis for representative members of plant and microbial SCOOP peptides is helpful and appreciated to deepen the discussions on their evolutionary aspects.

Our response: Thanks for the suggestion. We have included three phylogenetic trees, in which we compared the evolutionary relationships of SCOOPs in *Arabidopsis* (Supplementary Figure 2b), *Fusarium* SCOOPs (Supplementary Figure 7c), and bacterial SCOOPs (Supplementary Figure 9a). In addition, we have included all plant SCOOPs and microbial SCOOPs and constructed a phylogenetic tree with MEGAX using neighbor-joining methods (new Supplementary Figure 10b). We have included more discussions about the SCOOP/SCOOP evolution (the last paragraph of Discussion).

15) The mechanisms by which the plant PROSCOOPs or SCOOP peptides are released is not shown, and even whether this question is solved or unsolved is not mentioned in the text. Do they have an N-terminal secretion signal?

Our response: We thank this reviewer for the suggestion. In the revised manuscript, we have presented a schematic diagram showing putative SCOOP domains with an N-terminal signal peptide, variable regions and a conserved domain (new Supplementary Figure 2b) and the potential SCOOP processing from the precursor proteins (new Supplementary Fig. 3b). We have included new data showing that SCOOP12-GFP could be detected at the plasma membrane and extracellular spaces in *Arabidopsis pSCOOP12::SCOOP12-GFP/scoop12* transgenic plants (new Supplementary Figure 3c), suggesting that SCOOP12 is likely secreted. Further, SCOOP12 might be processed in plants as indicated by additional bands in immunoblots (new Supplementary Figure 3d). These data were described in Results (Page 6, Line 13-25).

References:

Harvey, K.L., Jarocki, V.M., Charles, I.G., and Djordjevic, S.P. (2019). The Diverse Functional Roles of Elongation Factor Tu (EF-Tu) in Microbial Pathogenesis. *Front Microbiol* 10, 2351.

He, Y., Wang, H., and Chen, L. (2015). Comparative secretomics reveals novel virulence-associated factors of *Vibrio parahaemolyticus*. *Front Microbiol* 6, 707.

Kunze, G., Zipfel, C., Robatzek, S., Niehaus, K., Boller, T., and Felix, G. (2004). The N terminus of bacterial elongation factor Tu elicits innate immunity in *Arabidopsis* plants. *Plant Cell* 16, 3496-3507.

Okuda, S., Fujita, S., Moretti, A., Hohmann, U., Doblas, V.G., Ma, Y., Pfister, A., Brandt, B., Geldner, N., and Hothorn, M. (2020). Molecular mechanism for the recognition of sequence-divergent CIF peptides by the plant receptor kinases GSO1/SGN3 and GSO2.

Proc Natl Acad Sci U S A. 117, 2693-2703.

Olsson, V., Joos, L., Zhu, S., Gevaert, K., Butenko, M.A., and De Smet, I. (2019). Look closely, the beautiful may be small: precursor-derived peptides in plants.

Annu Rev Plant Biol. 70, 153-186.

Santiago, J., Brandt, B., Wildhagen, M., Hohmann, U., Hothorn, L.A., Butenko, M.A., and Hothorn, M. (2016). Mechanistic insight into a peptide hormone signaling complex mediating floral organ abscission. *Elife* 5, e15075. doi: 10.7554/eLife.15075.

Sun, Y., Li, L., Macho, A.P., Han, Z., Hu, Z., Zipfel, C., Zhou, J.M., and Chai, J. (2013). Structural basis for flg22-induced activation of the *Arabidopsis* FLS2-BAK1 immune complex. *Science*. 342, 624-628.

Tang, J., Han, Z., Sun, Y., Zhang, H., Gong, X., and Chai, J. (2015). Structural basis for recognition of an endogenous peptide by the plant receptor kinase PEPR1. *Cell Res.* 25,

110-120.

Van der Does, D., Boutrot, F., Engelsdorf, T., Rhodes, J., McKenna, J.F., Vernhettes, S., Koevoets, I., Tintor, N., Veerabagu, M., Miedes, E., et al. (2017) The Arabidopsis leucine-rich repeat receptor kinase MIK2/LRR-KISS connects cell wall integrity sensing, root growth and response to abiotic and biotic stresses. *PLoS Genet.* 13, e1006832.

Wang, J., Li, H., Han, Z., Zhang, H., Wang, T., Lin, G., Chang, J., Yang, W., and Chai, J. (2015). Allosteric receptor activation by the plant peptide hormone phytosulfokine. *Nature* 525, 265-268.

REVIEWER COMMENTS

Reviewer #1 (Remarks to the Author):

The authors responded positively to my comments overall. However, it should be commented in the text of the manuscript (not just in the rebuttal letter) that the binding constants calculated from the binding experiments are significantly different from the effective concentrations estimated from the physiological experiments.

In addition, the *in vivo* structure of the SCOOP peptide, which was of concern also to other reviewers, was not directly evidenced in the revised manuscript. I understand that determination the mature structure of a secreted peptide requires a high degree of technical skill, but is it possible to determine the processing sites by immunoprecipitating the cleaved fragments of SCOOP12-GFP followed by MS analysis, for example? The following papers can be used as a reference for MS experiment (see Ni, J. et al. *Plant Mol Biol* 75, 67–75 (2011), Guo, Y. et al. *Plant Physiol.* 157, 476-484 (2011)).

Reviewer #2 (Remarks to the Author):

Comments

The authors did great job to address all our comments properly and added further data that supported their discoveries and strengthened their story in the manuscript. I only have a few comments for enhanced accuracy.

No free GFP was detected when SCOOP12-GFP was expressed in Suppl Fig 3d, in which a peptide containing conserved domain + C-term variable region is likely released rather than only conserved domain. The author should revise Suppl Fig 3b.

A model in Suppl Fig 10 is really confusing. Please simplify. For example, only one MIK2 receptor complex could be enough in the diagram. In addition, the diagram does not reflect that SCOOP12 plays a negative role in plant resistance against *E. amylovora* and *A. brassicola*. The diagram rather shows SCOOPs interfere these pathogens' infections. Since it did not addressed in the manuscript, I

recommend to delete the part. The same goes for *P. Syringae*. Instead, new arrows drawn from *Fusarium* and *Comamonadaceae* to the PRR complex can be added to represent MAMP from these pathogens.

Reviewer #3 (Remarks to the Author):

I appreciate very much the revisions and additional data provided by the authors.

There are still some concerns below, which need to be correctly addressed.

In Abstract, “a dual recognition capability” can be mistaken as a dual biochemical function recognizing two different ligand classes. This should be rephrased to “a dual function” or “a dual role” in sensing MAMPs and DAMPs, as stated on page 4.

Lines 26-28 on Page 3, it is not “generally believed” but it has been described, and this view is not merely based on MAMP-induced expression of DAMPs or their precursors. Studies cited in Ref 9 provide (at least genetic) evidence.

Line 18 page 5, "likes" should be "like".

Following Lines 18-23 page 5, the authors may wish to mention that DAMP peptide signaling often induces expression of themselves or their precursors, thereby providing a positive feedback for signal amplification. I assume this gave them a rationale to test whether MIK2 is required for SCOOP responses.

It is very good to show Suppl Fig 3, but the presence of a SCOOP12-GFP-derived band, which corresponds to free GFP in the immunoblot position, makes me wonder if the signals in Suppl Fig 3c represent SCOOP12, in particular in the apoplast. If the authors present the results, they also need to show a control image with free GFP.

Line 13 Page 7, it seems that Ca²⁺ and ROS bursts occurred mainly from shoots in the whole seedlings examined. The observed divergence among SCOOP peptide activities may rather reflect tissue

specificities. The two outputs do not necessarily represent “immune activation”. Careful wording is needed.

I appreciate the authors efforts to add RNA-seq results in Suppl Fig 4, but they do not seem to obey the standard. The analysis was done with two biological replicates and without receptor mutant or mock controls. Although not detailed in the method, the values at 1h and 6 h seem to be normalized with the values at 0 h. Cautions are needed in the interpretation of the data.

Which FocSCOOPL gene was knocked out in Fo5176? FOXG_11846? This needs to be clearly stated on Page 12. No information is provided for the structure and copy number (presence of homologues) of this gene in Fo5176. According to the apparent size of the gene in Suppl Fig 8a, this gene seems distinct from the one detailed in Suppl Fig 7 a and b. To assess the gene deletion effects (Suppl Fig 8b) and its knockout phenotype in plant inoculation assays (Suppl Fig 8 c-f), such information is also needed. The coding region of the gene does not seem large in size. If so, I am not sure if the authors’ argument (Fusarium SCOOPs reside in large proteins with unknown functions on page 16) is always the case.

In Suppl Fig 7c, are the SCOOPL sequences derived from the FGSG_07177 orthologues, or do they include that of another protein(s)? Is the SCOOPL sequence of FOXG_11846 in the alignment, or how does it look compared to these sequences? Whether SCOOPL sequences are seen in different proteins of Fo5176 needs to be described. If this is the case, it can be misleading to define this particular gene as “FocSCOOPL”.

The same concern also applies to Suppl Fig 9. Plant application assays are good, and I see that SCOOPL signature motifs are highly conserved in different microbes. However, it is not clear whether they are seen in different classes of proteins or different isoforms of the same kind (homologues) in these microbes.

A typo? “indivative in Line 31 page 15.

We again thank the editor and reviewers for their thorough reviews of our manuscript and additional insightful comments, to which we respond point-by-point below (in blue).

Reviewer #1 (Remarks to the Author):

The authors responded positively to my comments overall.

However, it should be commented in the text of the manuscript (not just in the rebuttal letter) that the binding constants calculated from the binding experiments are significantly different from the effective concentrations estimated from the physiological experiments.

Our response:

We thank the reviewer for the positive feedback on our revision. We have included this in the text (Page 10, Line 18-22). It reads as “It is noted that the binding constants calculated from the *in vitro* binding assays are significantly different from the effective concentrations estimated from the physiological experiments. The difference might be due to conditions between *in vivo* and *in vitro* assays and the involvement of other cellular components in regulating SCOOP12-MIK2 binding *in vivo* (Wang et al., 2015).”

In addition, the *in vivo* structure of the SCOOP peptide, which was of concern also to other reviewers, was not directly evidenced in the revised manuscript. I understand that determination the mature structure of a secreted peptide requires a high degree of technical skill, but is it possible to determine the processing sites by immunoprecipitating the cleaved fragments of SCOOP12-GFP followed by MS analysis, for example? The following papers can be used as a reference for MS experiment (see Ni, J. et al. *Plant Mol Biol* 75, 67–75 (2011), Guo, Y. et al. *Plant Physiol.* 157, 476-484 (2011)).

Our response:

We thank the reviewer for the suggestion. As suggested, we have followed the protocol (Guo et al., 2011; Ni et al., 2011) and determined the processing sites of SCOOP12 peptides via LC-MS/MS analysis. Our results indicated that the recombinant HIS-SCOOP12 proteins were processed upon incubation with the protein extracts of *Arabidopsis* seedlings (Supplementary Figure 3e). The processed SCOOP12 proteins were separated in an SDS-PAGE and subjected for in-gel digestion with endoproteinase Lys-C, followed by nano-flow LC-MS/MS (Nano-LC-MS/MS) analysis. A SCOOP12 fragment from leucine 44 (L⁴⁴) to lysine 78 (Y⁷⁸) was identified (Supplementary Figure 3f), suggesting that SCOOP12 was cleaved after arginine 43 (R⁴³) in a dibasic cleavage site (Supplementary Figure 3g). Importantly, when the dibasic residues of R⁴²R⁴³ were substituted with alanine residues (A⁴²A⁴³), one of the processed bands of SCOOP12-GFP expressed in protoplasts was abolished (Supplementary Figure 3h), further implicating the cleavage of SCOOP12 after R⁴³. The data are consistent with the predicted cleavage site between the variable region I and the conserved domain (Supplementary Figure 3c). These new data were described on Page 6-7, presented as Supplementary Figures 3e-h. The methods were detailed on Page 24-25.

Reviewer #2 (Remarks to the Author):

Comments

The authors did great job to address all our comments properly and added further data that supported their discoveries and strengthened their story in the manuscript. I only have a few comments for enhanced accuracy.

We thank the reviewer for the positive feedback on our revision.

No free GFP was detected when SCOOP12-GFP was expressed in Suppl Fig 3d, in which a

peptide containing conserved domain + C-term variable region is likely released rather than only conserved domain. The author should revise Suppl Fig 3b.

Our response:

Thanks for the suggestion. We agree with this reviewer that no free GFP was detected when SCOOP12-GFP was expressed in plants. In Supplementary Figure 3d, SCOOP12-GFP was detected as three major bands. We predicted that they might correspond to three fragments: 1) the variable region II + GFP, 2) the conserved domain + variable region II + GFP, and 3) the variable region I + conserved domain + variable region II+GFP. This is consistent with the schematic diagram in Supplementary Figure 3c. We also provided another repeat of Western blot to show the clear separation of different SCOOP12-GFP bands in new Supplementary Figure 3d. In addition, as suggested by Reviewer 1, we provided new data showing that the recombinant HIS-SCOOP12 proteins can be processed upon incubation with protein extracts of *Arabidopsis* seedlings and identified a cleavage site between arginine 43 (R43) and leucine 44 (L44) of SCOOP12 by LC-MS/MS analysis (Supplementary Figure 3e-h).

A model in Suppl Fig 10 is really confusing. Please simplify. For example, only one MIK2 receptor complex could be enough in the diagram. In addition, the diagram does not reflect that SCOOP12 plays a negative role in plant resistance against *E. amylovora* and *A. brassicola*. The diagram rather shows SCOOPs interfere these pathogens' infections. Since it did not address in the manuscript, I recommend to delete the part. The same goes for *P. Syringae*. Instead, new arrows drawn from *Fusarium* and *Comamonadaceae* to the PRR complex can be added to represent MAMP from these pathogens.

Our response:

Thanks for the suggestions. We have simplified the model as suggested. We have included only one MIK2 receptor complex in the diagram. We have deleted *E. amylovora*, *A. brassicola*, and *P. syringae*, and included new arrows from *Fusarium* and *Comamonadaceae* to the PRR complex in the model.

Reviewer #3 (Remarks to the Author):

I appreciate very much the revisions and additional data provided by the authors.

We thank the reviewer for the positive feedback on our revision.

There are still some concerns below, which need to be correctly addressed.

In Abstract, "a dual recognition capability" can be mistaken as a dual biochemical function recognizing two different ligand classes. This should be rephrased to "a dual function" or "a dual role" in sensing MAMPs and DAMPs, as stated on page 4.

Our response:

Thanks for the suggestion. We revised it as suggested.

Lines 26-28 on Page 3, it is not "generally believed" but it has been described, and this view is not merely based on MAMP-induced expression of DAMPs or their precursors. Studies cited in Ref 9 provide (at least genetic) evidence.

Our response:

Thanks for the suggestion. We have revised it to "Accumulating evidence indicates that....." in Page3, Line 26.

Line 18 page 5, "likes" should be "like".

Our response:

We changed it.

Following Lines 18-23 page 5, the authors may wish to mention that DAMP peptide signaling often induces expression of themselves or their precursors, thereby providing a positive feedback for signal amplification. I assume this gave them a rationale to test whether MIK2 is required for SCOOP responses.

Our response:

Thanks for the suggestion. We agree and have added this point in the revision (Page 5, Line 19-21).

It is very good to show Suppl Fig 3, but the presence of a SCOOP12-GFP-derived band, which corresponds to free GFP in the immunoblot position, makes me wonder if the signals in Suppl Fig 3c represent SCOOP12, in particular in the apoplast. If the authors present the results, they also need to show a control image with free GFP.

Our response:

Thanks for the suggestion. We have included a control image with free GFP in the new Supplementary Figure 3b. Our result indicates that, unlike SCOOP12-GFP, GFP was barely present in apoplasts. We also provided another Western blot to show a better separation of different SCOOP12-GFP bands in the new Supplementary Figure 3d.

Line 13 Page 7, it seems that Ca²⁺ and ROS bursts occurred mainly from shoots in the whole seedlings examined. The observed divergence among SCOOP peptide activities may rather reflect tissue specificities. The two outputs do not necessarily represent “immune activation”. Careful wording is needed.

Our response:

Thanks for the suggestion. We modified “immune activation” with “Ca²⁺ and ROS bursts” as suggested.

I appreciate the authors efforts to add RNA-seq results in Suppl Fig 4, but they do not seem to obey the standard. The analysis was done with two biological replicates and without receptor mutant or mock controls. Although not detailed in the method, the values at 1h and 6 h seem to be normalized with the values at 0 h. Cautions are needed in the interpretation of the data.

Our response:

We have performed a new RNA-Seq analysis of SCOOP12-regulated genes in the *mik2* mutant compared to WT plants. The data indicate that SCOOP12 treatment for 1 h regulated the expression of 2,249 genes (1,642 up-regulated and 607 down-regulated, fold change ≥ 2 and P-value ≤ 0.01) in WT plants compared to 24 regulated genes in *mik2*, suggesting that SCOOP12-regulated gene transcription is almost completely abolished in *mik2*. The data were described on Page 8-9, presented in Supplementary Figure 4f and Supplementary Table 2.

We also revised the Methods by adding “fold change of read counts at 1 and 6 h compared to that at 0 h”.

Which FocSCOOP gene was knocked out in Fo5176? FOXB_11846? This needs to be clearly stated on Page 12. No information is provided for the structure and copy number (presence of homologues) of this gene in Fo5176. According to the apparent size of the gene in Suppl Fig 8a, this gene seems distinct from the one detailed in Suppl Fig 7 a and b.

To assess the gene deletion effects (Suppl Fig 8b) and its knockout phenotype in plant inoculation assays (Suppl Fig 8 c-f), such information is also needed. The coding region of the gene does not seem large in size. If so, I am not sure if the authors’ argument (Fusarium SCOOPs reside in large proteins with unknown functions on page 16) is always the case.

Our response:

Thanks for bringing these points up. Yes, FocSCOOP1 is located in the N-terminus of FOXB_11846 in the Fo5176 strain. We knocked out the SCOOP1 sequence from the FOXB_11846 gene but not the whole gene. FOXB_11846, especially its SCOOP1 sequence, is conserved in all *Fusarium* species. This gene has only a single copy in each *Fusarium* species. We have added this information in the text and the figure legend (Page 12, Line 17-21; Page 42, Line 20-22; Page 43, Line 18).

In the original Supplementary Figure 8a, we only showed the SCOOP1 fragment, for which we replaced with the NEO gene, and flanking sequences of SCOOP1 used for gene recombination. We revised the diagram in the new Supplementary Figure 8a and showed the complete sequence of the FOXB_11846 gene harboring SCOOP1.

In Suppl Fig 7c, are the SCOOP1 sequences derived from the FGSG_07177 orthologues, or do they include that of another protein(s)? Is the SCOOP1 sequence of FOXB_11846 in the alignment, or how does it look compared to these sequences? Whether SCOOP1 sequences are seen in different proteins of Fo5176 needs to be described. If this is the case, it can be misleading to define this particular gene as “FocSCOOP1”.

Our response:

Yes, all the SCOOP1 sequences are derived from the FGSG_07177 orthologues in a variety of *Fusarium* species. FOXB_11846 is a FGSG_07177 orthologue in *F. oxysporum* f. sp. *conglutinans* strain 5176. The SCOOP1 sequence in FOXB_11846 of Fo5176 is included in the revised Supplementary Figure 7c.

We blast-searched SCOOP-like sequence using SCOOP12 and SCOOP10 as queries in *F. graminearum*, a *Fusarium* strain with a well-annotated genome sequence. Only one sequence highly similar to *Arabidopsis* SCOOPs was identified. We did a blast-search using the *F. graminearum* SCOOP1 sequence and found that SCOOP1 is conserved in FGSG_07177 orthologs in all searched *Fusarium* species. We explained the details in the text (Page 12, Line 17-21).

The same concern also applies to Suppl Fig 9. Plant application assays are good, and I see that SCOOP1 signature motifs are highly conserved in different microbes. However, it is not clear whether they are seen in different classes of proteins or different isoforms of the same kind (homologues) in these microbes.

Our response:

Similar to SCOOP1s in *Fusarium*, all the SCOOP1 sequences in Supplementary Figure 9a are derived from protein orthologs conserved in different genus or species of the *Comamonadaceae* family. We have included a new diagram in Supplementary Figure 9a to show the SCOOP1 position in the target protein. We also explained this in the text (Page 13, Line 19-20).

A typo? “indivative in Line 31 page 15.

Our response:

Thanks. We corrected this typo.

References

- Guo, Y., Ni, J., Denver, R., Wang, X., and Clark, S.E. (2011). Mechanisms of molecular mimicry of plant CLE peptide ligands by the parasitic nematode *Globodera rostochiensis*. *Plant Physiol* 157, 476-484.
- Gust, A.A., Pruitt, R., and Nurnberger, T. (2017). Sensing Danger: Key to Activating Plant Immunity. *Trends Plant Sci* 22, 779-791.
- Ni, J., Guo, Y., Jin, H., Hartsell, J., and Clark, S.E. (2011). Characterization of a CLE processing activity. *Plant Mol Biol* 75, 67-75.

Tanaka, K., and Heil, M. (2021). Damage-Associated Molecular Patterns (DAMPs) in Plant Innate Immunity: Applying the Danger Model and Evolutionary Perspectives. *Annu Rev Phytopathol.*

Wang, J., Li, H., Han, Z., Zhang, H., Wang, T., Lin, G., Chang, J., Yang, W., and Chai, J. (2015). Allosteric receptor activation by the plant peptide hormone phytosulfokine. *Nature* 525, 265-268.

REVIEWERS' COMMENTS

Reviewer #1 (Remarks to the Author):

My comment may not have been conveyed well, but my intention is to examine the N-terminal sequences of the SCOOP12-GFP fragments detected in Supplementary Fig. 3d. Sequence information will be obtained by immunoprecipitating the SCOOP12-GFP fragments in the sample with GFP antibody and performing LC-MS/MS analysis after enzymatic digestion. If the N-terminal sequence of each fragment is determined, the cleavage site can be predicted.

Reviewer #3 (Remarks to the Author):

I am happy with the revisions made by the authors. Great work.

One little concern is mutants' names. For instance, "pepr1,2" is used to represent pepr1 pepr2. Although it is correctly spelled in Methods, such abbreviations, if used, need to be defined at their first appearance in the main text.

Point-by-point response to reviewer comments

Reviewer #1 (Remarks to the Author):

My comment may not have been conveyed well, but my intention is to examine the N-terminal sequences of the SCOOP12-GFP fragments detected in Supplementary Fig. 3d. Sequence information will be obtained by immunoprecipitating the SCOOP12-GFP fragments in the sample with GFP antibody and performing LC-MS/MS analysis after enzymatic digestion. If the N-terminal sequence of each fragment is determined, the cleavage site can be predicted.

Our response:

We apologize for this. During the last revision, we have followed the protocol (Guo et al., 2011; Ni et al., 2011) suggested by the reviewer to determine the processing sites of SCOOP12 peptides via LC-MS/MS analysis. Guo et al., 2011 and Ni et al., 2011 also used the *E. coli* purified peptide proteins incubated with plant extracts. It is technically challenging to obtain enough immunoprecipitated SCOOP12-GFP fragments from transgenic plants for cleavage site identification. As suggested by the editor, we have noted that the predicted cleavage sites derived from the *in vitro* assay using plant extracts might not be fully representative of the situation *in vivo* (Page 7 Line 10-12).

Reviewer #3 (Remarks to the Author):

I am happy with the revisions made by the authors. Great work.

One little concern is mutants' names. For instance, "pepr1,2" is used to represent pepr1 pepr2. Although it is correctly spelled in Methods, such abbreviations, if used, need to be defined at their first appearance in the main text.

Our response:

Thanks for the suggestion. We have checked all mutants' names and made all these names be correctly spelled and abbreviations have been defined at their first appearance in the main text.